# IKKε and TBK1 prevent RIPK1 dependent and independent inflammation

Remzi Onur Eren[1], Göksu Gökberk Kaya ®[1], Robin Schwarzer[1,3] &
Manolis Pasparakis ®[1,2] ✉

TBK1 and IKKε regulate multiple cellular processes including anti-viral type-I interferon responses, metabolism and TNF receptor signaling. However, the relative contributions and potentially redundant functions of IKKε and TBK1 in cell death, inflammation and tissue homeostasis remain poorly understood. Here we show that IKKε compensates for the loss of TBK1 kinase activity to prevent RIPK1-dependent and -independent inflammation in mice. Combined inhibition of IKKε and TBK1 kinase activities caused embryonic lethality that was rescued by heterozygous expression of kinase-inactive RIPK1. Adult mice expressing kinase-inactive versions of IKKε and TBK1 developed systemic inflammation that was induced by both RIPK1-dependent and -independent mechanisms. Combined inhibition of IKKε and TBK1 kinase activities in myeloid cells induced RIPK1-dependent cell death and systemic inflammation mediated by IL-1 family cytokines. Tissue-specific studies showed that IKKε and TBK1 were required to prevent cell death and inflammation in the intestine but were dispensable for liver and skin homeostasis. Together, these findings revealed that IKKε and TBK1 exhibit tissue-specific functions that are important to prevent cell death and inflammation and maintain tissue homeostasis.

Recognition of both self and non-self-nucleic acids by innate immune receptors triggers potent activation of type I interferon (IFN) responses[1]. TANK binding kinase 1 (TBK1) induces type I IFN responses downstream of nearly all innate nucleic acid sensors. Toll-like receptor (TLR)−3 and −4, the cytosolic RNA sensors RIG-I and MDA5 as well as the cytosolic DNA sensor cyclic GMP-AMP synthase (cGAS) utilize the adapter proteins TRIF, MAVS and STING, respectively, to induce TBK1-mediated type I IFN activation[2–10]. Upon ligand engagement by the respective receptors, TBK1 phosphorylates these adaptor proteins resulting in the recruitment of the transcription factor interferon regulatory factor 3 (IRF3)[11]. Subsequently, TBK1 phosphorylates IRF3 leading to its homo-dimerization and nuclear translocation, culminating in the transcriptional activation of type I IFNs and interferon-stimulated genes (ISGs)[12–14].

TBK1 is also involved in tumor necrosis factor receptor 1 (TNFR1) signaling. Binding of TNF to TNFR1 induces receptor trimerization and

recruitment of the adaptor protein TRADD (TNF receptor type 1-associated death domain protein) and receptor-interacting protein kinase 1 (RIPK1)[15], which nucleate the formation of a receptor proximal signaling complex (termed complex I) including the ubiquitin ligases cIAP1/2 and LUBAC. Ubiquitination of RIPK1 and other components of the complex with lysine 63-linked and linear ubiquitin chains induces the recruitment of TAK1 and the IKK complex, which trigger the activation of the transcription factor NF-κB and the induction of proinflammatory and pro-survival genes. Inhibition of NF-κB-dependent pro-survival gene expression, destabilization of complex I by inhibition of cIAPs and LUBAC or inhibition of TAK1 or IκB kinases (IKKs) result in the formation of cytosolic cell death-inducing protein complexes, which recruit and activate caspase-8 through the adapter FADD resulting in apoptosis. When caspase-8 is inhibited, TNFR1 induces necroptosis, a necrotic type of cell death mediated by RIPK3, which phosphorylates the pseudokinase MLKL triggering its oligomerization

[1]Institute for Genetics and Cologne Excellence Cluster on Cellular Stress Responses in Aging-Associated Diseases (CECAD), University of Cologne, Cologne, Germany. [2]Center for Molecular Medicine (CMMC), University of Cologne, Cologne, Germany. [3]Present address: Genentech Inc, South San Francisco, USA. ✉e-mail: pasparakis@uni-koeln.de

and translocation to the plasma membrane inducing cell lysis[16]. RIPK1 determines the outcome of TNFR1 signaling, as it acts in a kinase-independent manner within complex I to induce NF-κB activation, whereas activation of RIPK1 kinase activity triggers caspase-8-dependent apoptosis or RIPK3-MLKL-dependent necroptosis[17,18]. Phosphorylation of RIPK1 by IKK1 and IKK2[19,20], as well as MK2[21–23] prevents its activation and inhibits TNF-induced cell death. TBK1-deficient mice died during development but could be rescued by TNFR1 or TNF deficiency[24,25], demonstrating an important role of TBK1 in preventing TNF-induced embryonic lethality. Moreover, the embryonic lethal phenotype of $Tbk1^{-/-}$ mice was also rescued by crossing with mice expressing kinase-inactive RIPK1 ($Ripk1^{D138N/D138N}$)[26], showing that TBK1 prevents RIPK1-dependent cell death. Importantly, mutations resulting in loss of TBK1 expression in humans did not result in embryonic lethality but caused autoinflammatory pathology that was effectively treated with TNF-blocking antibodies[27]. These studies provided evidence that TBK1 has a critical role in preventing TNFR1-mediated and RIPK1-dependent cell death.

IKKε is homologous to TBK1 and was also shown to contribute to IFN production, with the two kinases exhibiting functional redundancy in anti-viral responses[25,27]. IKKε was also implicated in regulating TNFR1 and RIPK1 signaling together with TBK1. Specifically, both IKKε and TBK1 were recruited to the TNFR1 signaling complex and were shown to phosphorylate RIPK1 to inhibit its activation and the induction of cell death[28]. However, in contrast to $Tbk1^{-/-}$ mice that showed TNF-dependent embryonic lethality, IKKε-deficient mice developed normally and did not show apparent pathology, demonstrating that IKKε is dispensable for embryonic development and normal tissue homeostasis[2]. Interestingly, mice lacking both IKKε and TBK1 died during embryogenesis similarly to $Tbk1^{-/-}$, however, they could not be rescued by TNF deficiency[25], showing that combined loss of IKKε and TBK1 causes embryonic lethality by mechanisms that are at least in part independent of TNF. Thus, whereas IKKε appears to share important functions with TBK1, the combined role of these two kinases in regulating embryonic development and tissue homeostasis and the underlying mechanisms remain elusive. Here we investigated the role of TBK1 and IKKε by generating and analyzing knock-in mice expressing kinase-inactive versions of these kinases. Our results revealed a previously unappreciated function of IKKε in compensating for the loss of TBK1 kinase activity to prevent RIPK1-dependent and -independent cell death and inflammation, which is important for the maintenance of tissue homeostasis.

## Results

### Combined inhibition of IKKε and TBK1 caused RIPK1 kinase activity-dependent embryonic lethality and transient alopecia

To study the physiological role of the kinase activity of TBK1 and IKKε in vivo, we generated mice expressing catalytically inactive versions of these kinases. Specifically, we introduced mutations substituting aspartic acid at position 135 of TBK1 with arginine (TBK1-D135N) and lysine at position 38 of IKKε with alanine (IKKε-K38A) in the respective endogenous genes using CRISPR/Cas9-mediated gene targeting (Fig. 1a, b). To compare the loss of kinase activity to the complete lack of protein expression, we also generated mice lacking TBK1 expression by crossing mice carrying loxP-flanked $Tbk1$ alleles[29] ($Tbk1^{fl/fl}$) to Deleter-Cre mice[30], as well as mice lacking IKKε by using CRISPR/Cas9-mediated gene targeting to introduce a frameshift mutation in the $Ikke$ gene ($Ikke^{-/-}$) (Fig. 1a, b). $Ikke^{-/-}$ and $Ikke^{K38A/K38A}$ mice were viable and fertile and did not show any apparent pathology consistent with earlier studies in $Ikke^{-/-}$ mice[2]. $Tbk1^{-/-}$ mice died during embryogenesis but were rescued by homozygous or heterozygous expression of kinase-inactive RIPK1-D138N (Fig. 1c), as reported previously[26]. $Tbk1^{D135N/D135N}$ mice were also embryonically lethal and were rescued by homozygous or heterozygous RIPK1-D138N expression (Fig. 1c), showing that lack of TBK1 catalytic activity caused RIPK1 kinase-dependent embryonic lethality.

To assess potential redundant functions between TBK1 and IKKε, we investigated whether combined loss of these kinases or their catalytic activity also causes RIPK1 kinase-dependent embryonic lethality. We found that $Ikke^{-/-}$ $Tbk1^{-/-}$ and $Ikke^{K38A/K38A}$ $Tbk1^{D135N/D135N}$ were also embryonically lethal but were born and reached adulthood when RIPK1 kinase activity was inhibited by heterozygous or homozygous expression of catalytically inactive RIPK1-D138N (Fig. 1d). Immunoblot analysis of bone marrow-derived macrophages (BMDMs) revealed that the kinase-inactive IKKε-K38A and TBK1-D135N proteins were expressed at normal levels and that loss of kinase activity or protein expression of TBK1 alone or together with IKKε prevented the phosphorylation of IRF3 in response to stimulation with lipopolysaccharide (LPS) (Fig. 1e). Interestingly, we found that $Ikke^{-/-}$ $Tbk1^{-/-}$ $Ripk1^{wt/D138N}$ and $Ikke^{K38A/K38A}$ $Tbk1^{D135N/D135N}$ $Ripk1^{wt/D138N}$, but not $Tbk1^{-/-}$ $Ripk1^{wt/D138N}$ or $Tbk1^{D135N/D135N}$ $Ripk1^{wt/D138N}$ mice, displayed alopecia before weaning with hair growth recovering almost completely by the age of 8 weeks (Fig. 1f, g). This hair phenotype depended on RIPK1 kinase activity as $Ikke^{-/-}$ $Tbk1^{-/-}$ $Ripk1^{D138N/D138N}$ and $Ikke^{K38A/K38A}$ $Tbk1^{D135N/D135N}$ $Ripk1^{D138N/D138N}$ did not show transient alopecia (Fig. 1f, g). To assess whether the transient alopecia was caused by loss of a keratinocyte-intrinsic function of IKKε and TBK1, we generated $Ikke^{K38A/K38A}$ $Tbk1^{fl/D135N}$ $K14$-Cre$^{wt/tg}$ mice that lack TBK1 kinase activity specifically in keratinocytes and IKKε kinase activity in all cells. $Ikke^{K38A/K38A}$ $Tbk1^{fl/D135N}$ $K14$-Cre$^{wt/tg}$ mice did not develop alopecia (Fig. 1f, g), showing that combined inhibition of IKKε and TBK1 kinase activity caused RIPK1 kinase-dependent transient hair loss by a non-keratinocyte-intrinsic mechanism.

### Systemic or myeloid cell-specific inhibition of TBK1 and IKKε causes granulocytosis and monocytosis

On necropsy, $Ikke^{-/-}$ $Tbk1^{-/-}$ $Ripk1^{wt/D138N}$ and $Ikke^{K38A/K38A}$ $Tbk1^{D135N/D135N}$ $Ripk1^{wt/D138N}$ mice, but not $Tbk1^{-/-}$ $Ripk1^{wt/D138N}$ and $Tbk1^{D135N/D135N}$ $Ripk1^{wt/D138N}$ mice, displayed splenomegaly, which was partially suppressed by homozygous $Ripk1^{D138N/D138N}$ genetic background (Fig. 2a). Immunophenotyping using flow cytometry (Supplementary Fig. 1a) revealed that 8-13 week-old $Ikke^{-/-}$ $Tbk1^{-/-}$ $Ripk1^{wt/D138N}$ and $Ikke^{K38A/K38A}$ $Tbk1^{D135N/D135N}$ $Ripk1^{wt/D138N}$ mice had increased numbers of monocytes and granulocytes in peripheral blood, enlarged spleens that contained an elevated percentage of neutrophils (CD11b+ Ly6G+) and a pronounced increase in the percentage of monocytes (CD11b+ CD115+) in the bone marrow (Fig. 2b–d). This phenotype depended mainly on RIPK1 kinase activity as a homozygous expression of RIPK1-D138N largely normalized the amount of monocytes and neutrophils (Fig. 2b–d). Importantly, $Tbk1^{-/-}$ $Ripk1^{wt/D138N}$ and $Tbk1^{D135N/D135N}$ $Ripk1^{wt/D138N}$ mice did not show this phenotype, arguing that IKKε kinase activity suppresses RIPK1 kinase-dependent monocytosis and neutrophilia development in the absence of TBK1 kinase activity. We did not observe considerable differences in splenic B and T cell and thymocyte populations (Supplementary Fig. 1b–e). Mice with combined deficiency or kinase inhibition of IKKε and TBK1 on $Ripk1^{wt/D138N}$ or $Ripk1^{D138N/D138N}$ genetic background displayed an activated T cell signature in the spleen indicated by increased CD69 to CD62L ratio (Fig. 2e). Collectively, these results revealed that combined inhibition of IKKε and TBK1 kinase activities caused myeloid cell expansion and T cell activation that were mediated by RIPK1 kinase activity-dependent and -independent pathways.

To investigate whether the granulocytosis and monocytosis observed in $Ikke^{K38A/K38A}$ $Tbk1^{D135N/D135N}$ $Ripk1^{wt/D138N}$ mice were caused by loss of myeloid cell-intrinsic function of TBK1 and IKKε, we generated $Ikke^{K38A/K38A}$ $Tbk1^{fl/D135N}$ $Cx3cr1$-Cre$^{wt/tg}$ mice, which lack TBK1 kinase activity in myeloid cells and IKKε kinase activity in all cells (Fig. 3a). We found that $Ikke^{K38A/K38A}$ $Tbk1^{fl/D135N}$ $Cx3cr1$-Cre$^{wt/tg}$ mice developed splenomegaly, granulocytosis, and monocytosis that were not observed in $Tbk1^{fl/D135N}$ $Cx3cr1$-Cre$^{wt/tg}$ mice (Fig. 3b–e). Moreover, $Ikke^{K38A/K38A}$ $Tbk1^{fl/D135N}$ $Cx3cr1$-Cre$^{wt/tg}$ mice showed a relative decrease in the percentage but not total number of splenic CD4+ and CD8 + T cells, which

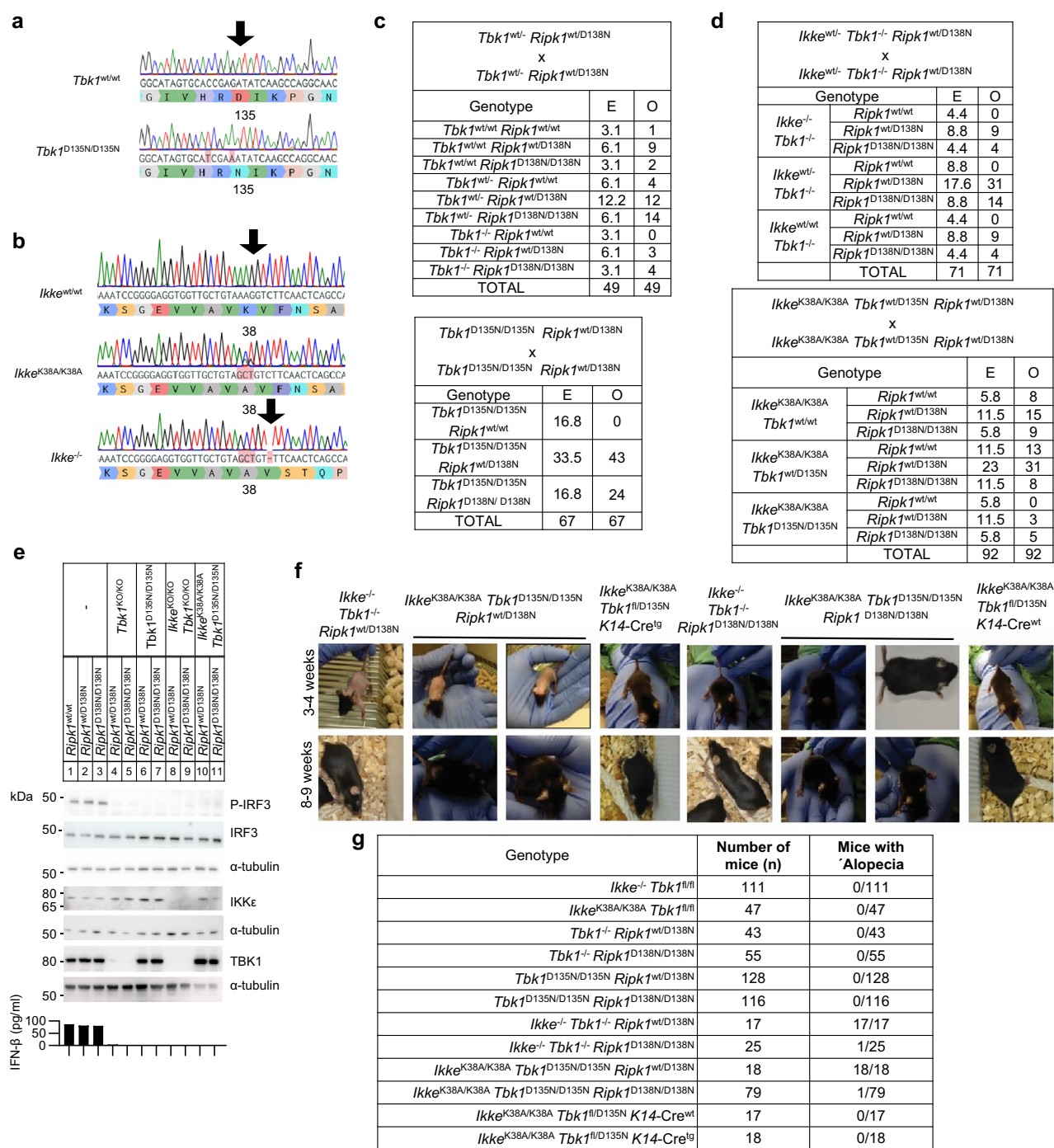

**Fig. 1 | Combined deficiency or kinase inhibition of TBK1 and IKKε cause RIPK1-dependent embryonic lethality and transient alopecia. a, b** Generation of mice with *Tbk1*^D135N/D135N **a**, *Ikke*^K38A/K38A and *Ikke*^-/- **b** alleles using CRISPR/Cas9-mediated genome editing. Sanger DNA sequencing of genomic DNA confirmed the generation of the correctly targeted alleles. **c, d** Expected (E) and observed (O) numbers of weaned offspring with the indicated genotypes from intercrosses of parents with the indicated genotypes. **e** Immunoblot analysis with the indicated antibodies of whole cell lysates from BMDMs from mice with the indicated genotypes stimulated for 1 hour with 100 ng/ml LPS (top), and ELISA analysis of IFN-β levels in the supernatant of BMDMs from indicated genotypes for 24 h with 100 ng/ml LPS (bottom). Data shown are representative of two independent experiments. **f** Representative pictures of mice with the indicated age and genotype. **g** Occurrence of alopecia phenotype in mice with the indicated genotype at pre-weaning age. Source data for **e** is provided as a source data file.

had an activated phenotype indicated by an increased CD69 to CD62L ratio (Fig. 3f, g). Importantly, homozygous expression of RIPK1-D138N prevented both the splenomegaly and expansion of myeloid cells as well as the activated T cell phenotype in *Ikke*^K38A/K38A *Tbk1*^fl/D135N *Cx3cr1*-Cre^wt/tg *Ripk1*^D138N/D138N mice (Fig. 3b–g). Together, these results showed that IKKε and TBK1 share a redundant myeloid cell-intrinsic function that is essential to prevent RIPK1 kinase-dependent granulocytosis, monocytosis and T cell activation.

## Combined inhibition of IKKε and TBK1 caused RIPK1-dependent liver inflammation

Histological analysis of liver sections revealed the presence of apoptotic cells, identified by immunostaining with antibodies against cleaved caspase-8, as well as increased infiltration of CD45^+ immune cells in livers from *Ikke*^-/- *Tbk1*^-/- *Ripk1*^wt/D138N and *Ikke*^K38A/K38A *Tbk1*^D135N/D135N *Ripk1*^wt/D138N mice (Fig. 4a). Furthermore, *Ikke*^-/- *Tbk1*^-/- *Ripk1*^wt/D138N and *Ikke*^K38A/K38A *Tbk1*^D135N/D135N *Ripk1*^wt/D138N mice had elevated amounts of

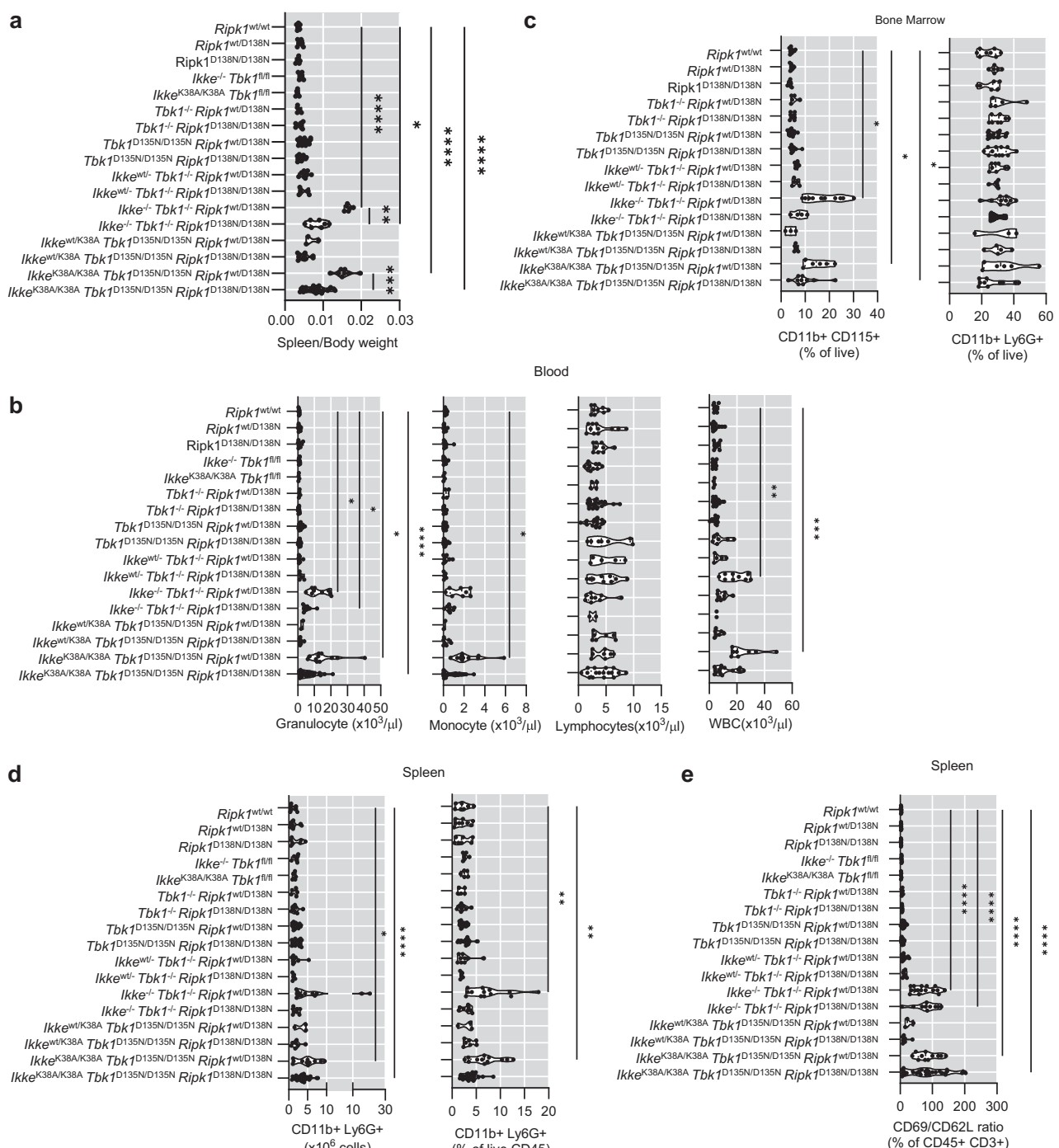

**Fig. 2 | Combined deficiency or kinase inhibition of TBK1 and IKKε cause RIPK1-dependent and -independent systemic inflammation. a**, **b** Graphs showing spleen to body weight ratio **a** and peripheral blood analysis assayed by differential hematology analyzer **b** of 8-13 week-old mice with the indicated genotypes. **c**, **d** Graphs showing flow cytometry analysis of CD11b + CD115+ (monocytes) and CD11b + Ly6G+ (neutrophils) in bone marrow **c** and spleen **d** of 8–13 week-old mice with the indicated genotypes. **e** Graphs showing ratio of CD69+ (activated) to CD62L+ (naive) T cells amongst live CD45 + CD3+ splenocytes of 8–13 weeks old mice week-old mice with the indicated genotypes. Each dot represents one mouse. $^*p < 0.05$, $^{**}p < 0.01$, $^{***}p < 0.005$, $^{****}p < 0.0001$ (one-way ANOVA t-test with post-hoc multiple test). Violin plots show the median of biological replicates and interquartile range of data. Source data for **a**–**e** are provided as a source data file.

serum alanine aminotransferase (ALT), alkaline phosphatase (ALP) and aspartate aminotransferase (AST), markers of liver damage, compared to controls as well as *Tbk1*[−/−] *Ripk1*[wt/D138N] and *Tbk1*[D135N/D135N] *Ripk1*[wt/D138N] mice (Fig. 4b, c), showing that systemic inhibition of IKKε and TBK1 kinase activity caused liver pathology. Importantly, homozygous expression of RIPK1-D138N normalized markers of liver damage and strongly reduced the presence of apoptotic cells and infiltrating immune cells in livers from *Ikke*[−/−] *Tbk1*[−/−] *Ripk1*[D138N/D138N] and *Ikke*[K38A/K38A]

*Tbk1*[D135N/D135N] *Ripk1*[D138N/D138N] mice (Fig. 4a–c), showing that combined inhibition of IKKε and TBK1 caused liver damage mediated predominantly by RIPK1 kinase activity-dependent signaling.

The liver pathology caused by combined inhibition of IKKε and TBK1 could be due to a hepatocyte-intrinsic function or an indirect effect of the myeloid cell expansion observed in these animals. Interestingly, *Ikke*[K38A/K38A] *Tbk1*[fl/D135N] *Cx3cr1*-Cre[wt/tg] mice showed increased infiltration of immune cells and the presence of small numbers of

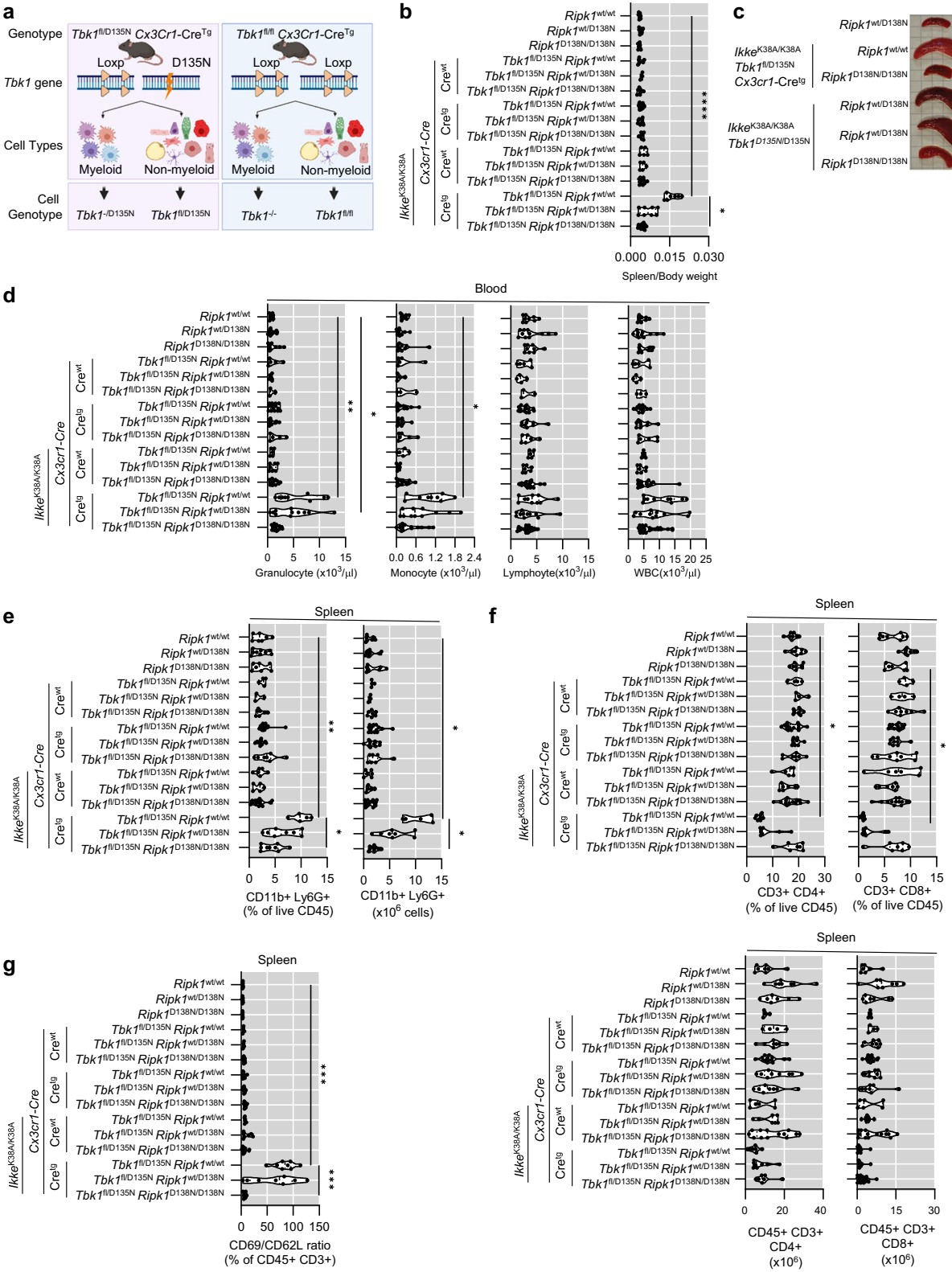

**Fig. 3 | Combined inhibition of TBK1 and IKKε kinase activities specifically in myeloid cells causes RIPK1 kinase activity-dependent systemic inflammation. a** Schematic depicting the breeding strategy used to generate mice with combined inhibition of TBK1 and IKKε kinase activities in myeloid cells. Created with BioRender.com. **b, c** Graph showing spleen to body weight ratio **b** and representative pictures of spleens **c** of 8–13 week-old mice with the indicated genotypes. **d** Graphs depicting peripheral blood analysis of mice with the indicated genotypes, assayed by differential hematology analyzer. **e, f** Graphs showing cell counts and

percentage of CD11b + Ly6G+ (neutrophils) **e**, as well as CD3 + CD4+ and CD3 + CD8+ cells **f** among live and CD45+ splenocytes of 8–13 week-old mice with the indicated genotypes. **g** Graphs showing the ratio of CD69+ (activated) to CD62L+ (naive) T cells amongst live CD45 + CD3+ splenocytes of 8-13 week-old mice with indicated genotypes. Each dot represents one mouse. $^*p < 0.05$, $^{**}p < 0.01$, $^{***}p < 0.005$, $^{****}p < 0.0001$ (one-way ANOVA t-test with post-hoc multiple test). Violin plots show the median of biological replicates and interquartile range of data. Source data for **b** and **d–g** are provided as a source data file.

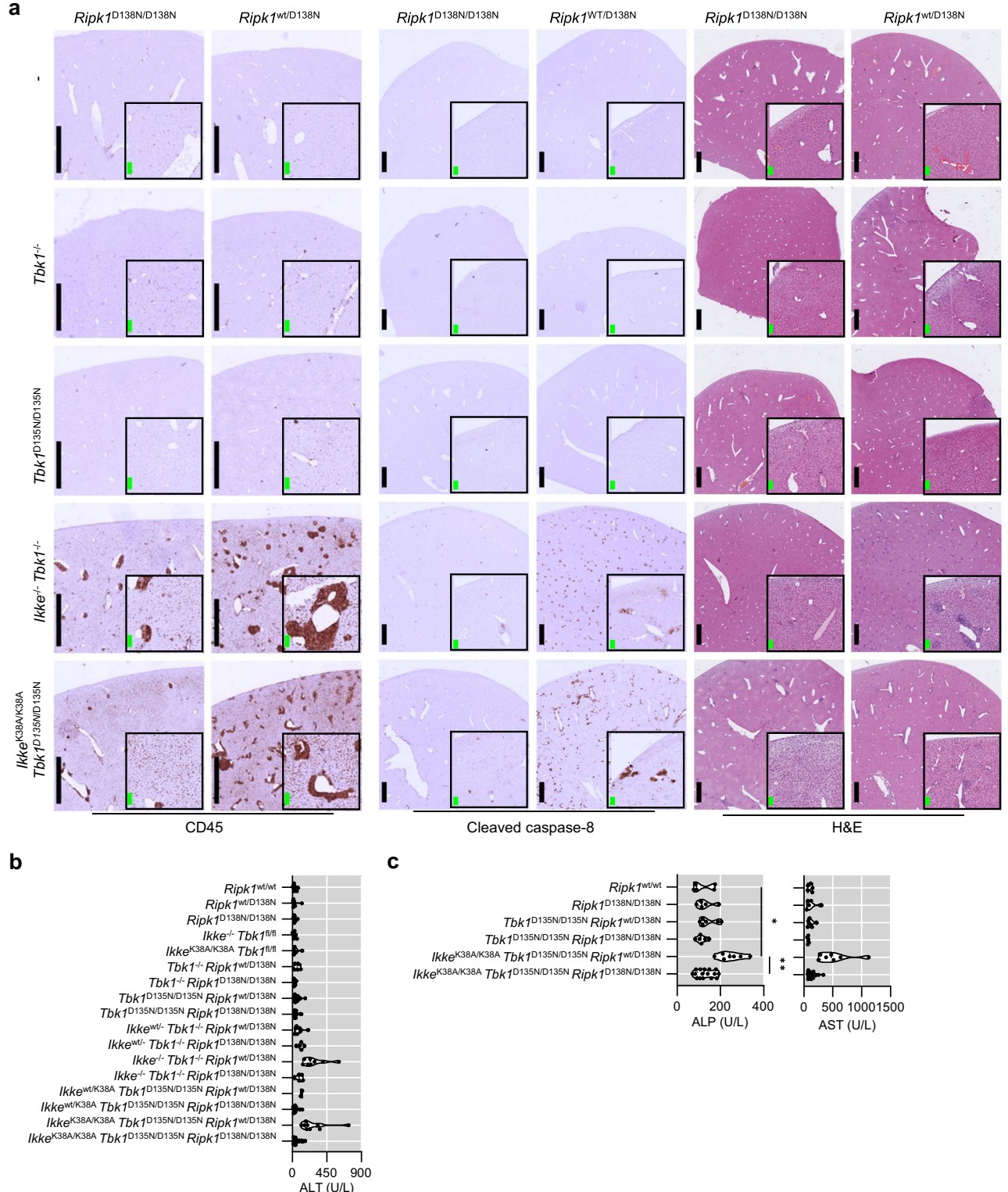

**Fig. 4 | Combined systemic inhibition of TBK1 and IKKε kinase activities causes RIPK1-kinase activity dependent liver damage and inflammation.**
**a** Representative images of liver sections from 8-13 week-old mice with the indicated genotypes immunostained for CD45 and cleaved caspase 8 or stained with hematoxylin and eosin (H&E). Scale bars, 1 mm (black) and 100 μm (green).

**b, c** Serum ALT, ALP and AST levels of 8-13-week-old mice with the indicated genotypes. Each dot represents one mouse. $^*p < 0.05$, $^{**}p < 0.01$, $^{***}p < 0.005$, $^{****}p < 0.0001$. (one-way ANOVA t-test with post-hoc multiple test). Violin plots show the median of biological replicates and interquartile range of data. Source data for **b, c** are provided as a source data file.

cleaved caspase-8 positive cells in their livers, which depended on RIPK1 kinase activity (Fig. 5a). However, $Ikke^{K38A/K38A}$ $Tbk1^{fl/D135N}$ $Cx3cr1$-$Cre^{wt/tg}$ mice had normal serum ALT values showing that despite the small number of dying cells and mild liver inflammation, myeloid cell specific inhibition of TBK1 and IKKε was not sufficient to cause considerable liver damage (Fig. 5b). To address whether hepatocyte-intrinsic inhibition of IKKε and TBK1 caused the liver pathology, we generated $Ikke^{K38A/K38A}$ $Tbk1^{fl/D135N}$ $Afp$-$Cre^{wt/tg}$ mice, which lack TBK1

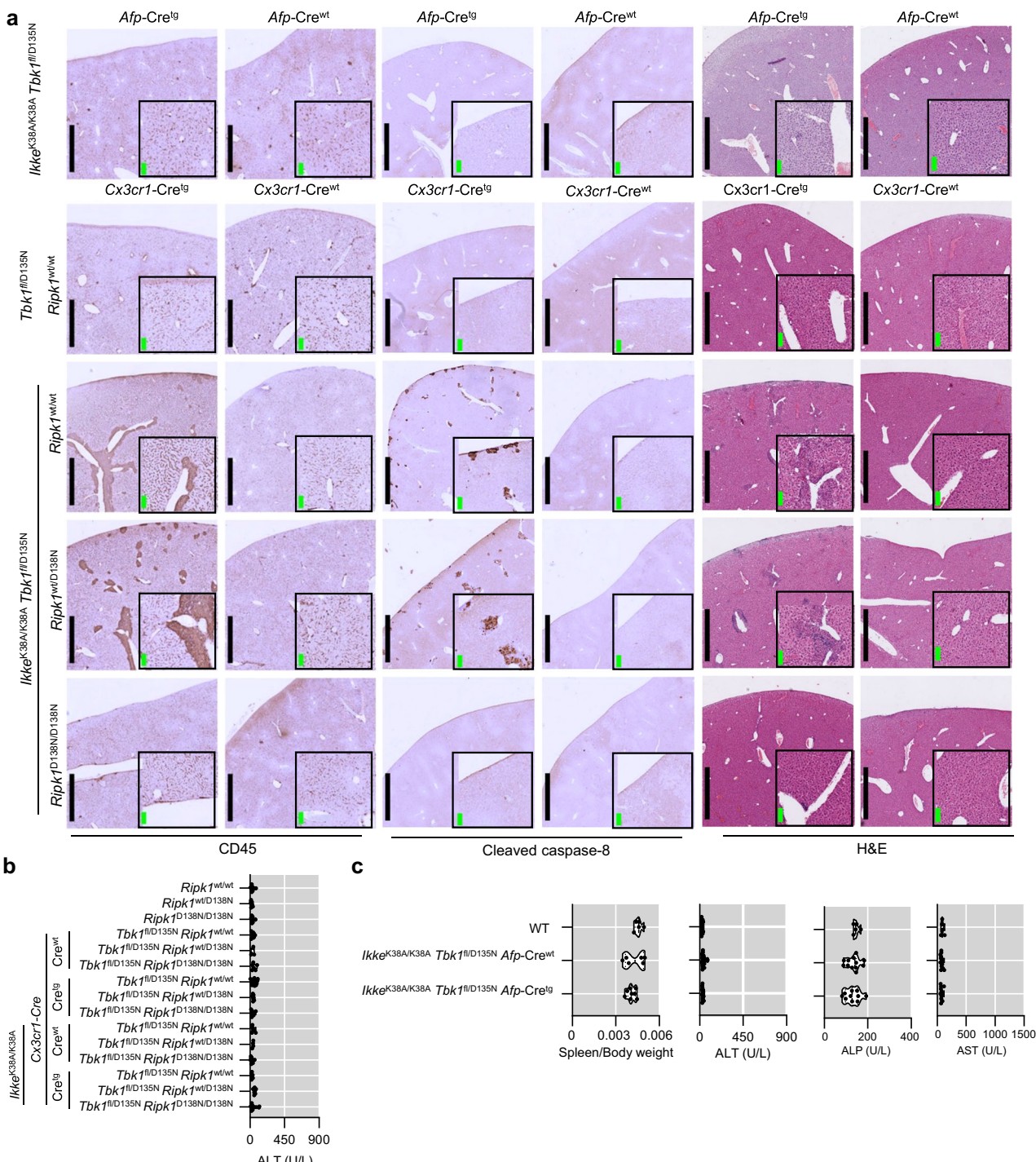

**Fig. 5 | Combined inhibition of TBK1 and IKKε kinase activities specifically in myeloid cells, but not in liver parenchymal cells, causes RIPK1-dependent liver inflammation. a** Representative images of liver sections from 8 to 13 week-old mice with the indicated genotypes immunostained for CD45 and cleaved caspase 8 or stained with hematoxylin and eosin (H&E). Scale bars, 1 mm (black) and 100 μm (green). **b, c** Serum ALT, ALP, and AST levels of 8-13-week-old mice with the indicated genotypes. Each dot represents one mouse. $*p < 0.05$, $**p < 0.01$, $***p < 0.005$, $****p < 0.0001$. (one-way ANOVA t-test with post-hoc multiple test). Violin plots show the median of biological replicates and interquartile range of data. Source data for **b** and **c** are provided as a source data file.

kinase activity specifically in liver parenchymal cells (LPCs) and IKKε kinase activity in all cells. *Ikke*[K38A/K38A] *Tbk1*[fl/D135N] *Afp*-Cre[wt/tg] mice displayed normal spleen size and did not show liver pathology as judged by the absence of apoptotic cells and infiltrating immune cells as well as normal values of serum ALT, ALP and AST (Fig. 5a, c). Therefore, neither myeloid cell-specific nor liver parenchymal cell-specific IKKε and TBK1 inhibition were sufficient to cause liver damage, suggesting

that combined inhibition of these kinases in both immune and parenchymal cells triggered cell death and inflammation in the liver.

## IKKε and TBK1 function in both myeloid and epithelial cells to maintain intestinal immune homeostasis

To assess whether IKKε and TBK1 inhibition affects intestinal homeostasis, we performed immunohistological analysis of sections from

the ileum and colon. We found that both *Ikke*[-/-] *Tbk1*[-/-] *Ripk1*[wt/D138N] and *Ikke*[K38A/K38A] *Tbk1*[D135N/D135N] *Ripk1*[wt/D138N] mice showed increased immune cell infiltration accompanied by the presence of elevated numbers of apoptotic cells identified by immunostaining for cleaved caspase-3 in the ileum, which was partially prevented by homozygous RIPK1-D138N expression (Fig. 6a–c). The colon of these mice was less affected with mild immune cell infiltration and slightly increased numbers of dying cells (Fig. 6a–c). To assess whether the intestinal phenotype was caused by the inhibition of TBK1 and IKKε in myeloid cells, we analyzed intestinal tissues from *Tbk1*[fl/D135N] *Cx3cr1*-Cre[wt/tg] and *Ikke*[K38A/K38A] *Tbk1*[fl/D135N] *Cx3cr1*-Cre[wt/tg] mice. We found that *Ikke*[K38A/K38A] *Tbk1*[fl/D135N] *Cx3cr1*-Cre[wt/tg] but not *Tbk1*[fl/D135N] *Cx3cr1*-Cre[wt/tg] mice showed increased immune cell infiltration in both the ileum and colon, which were strongly suppressed by homozygous expression of RIPK1-D138N (Fig. 7a, b). Moreover, sections from both the colon and small intestine of *Ikke*[K38A/K38A] *Tbk1*[fl/D135N] *Cx3cr1*-Cre[wt/tg] mice had

increased numbers of cleaved caspase-3 positive cells, which were reduced by inhibition of RIPK1 kinase activity (Data Fig. 7a, c). Furthermore, we observed that *Ikke*[K38A/K38A] mice lacking TBK1 kinase activity systemically or specifically in myeloid cells had decreased numbers of Goblet cells identified by Alcian Blue staining in the terminal ileum and, to some extent, in the colon, a phenotype that was only partially dependent on RIPK1 kinase activity (Supplementary Fig. 2).

To dissect the contribution of intestinal epithelial cell-intrinsic functions of IKKε and TBK1, we generated mice lacking specifically TBK1-kinase activity in intestinal epithelial cells (IECs) on *Ikke*[wt/K38A] and *Ikke*[K38A/K38A] background. We found that *Ikke*[K38A/K38A] *Tbk1*[fl/D135N] *Villin*-Cre[wt/tg], but not *Ikke*[wt/K38A] *Tbk1*[fl/D135N] *Villin*-Cre[wt/tg] mice, showed reduced body weight and displayed intestinal pathology characterized by epithelial hyperplasia, increased numbers of dying cells as well as a marked reduction of secretory cells in the terminal ileum and to a

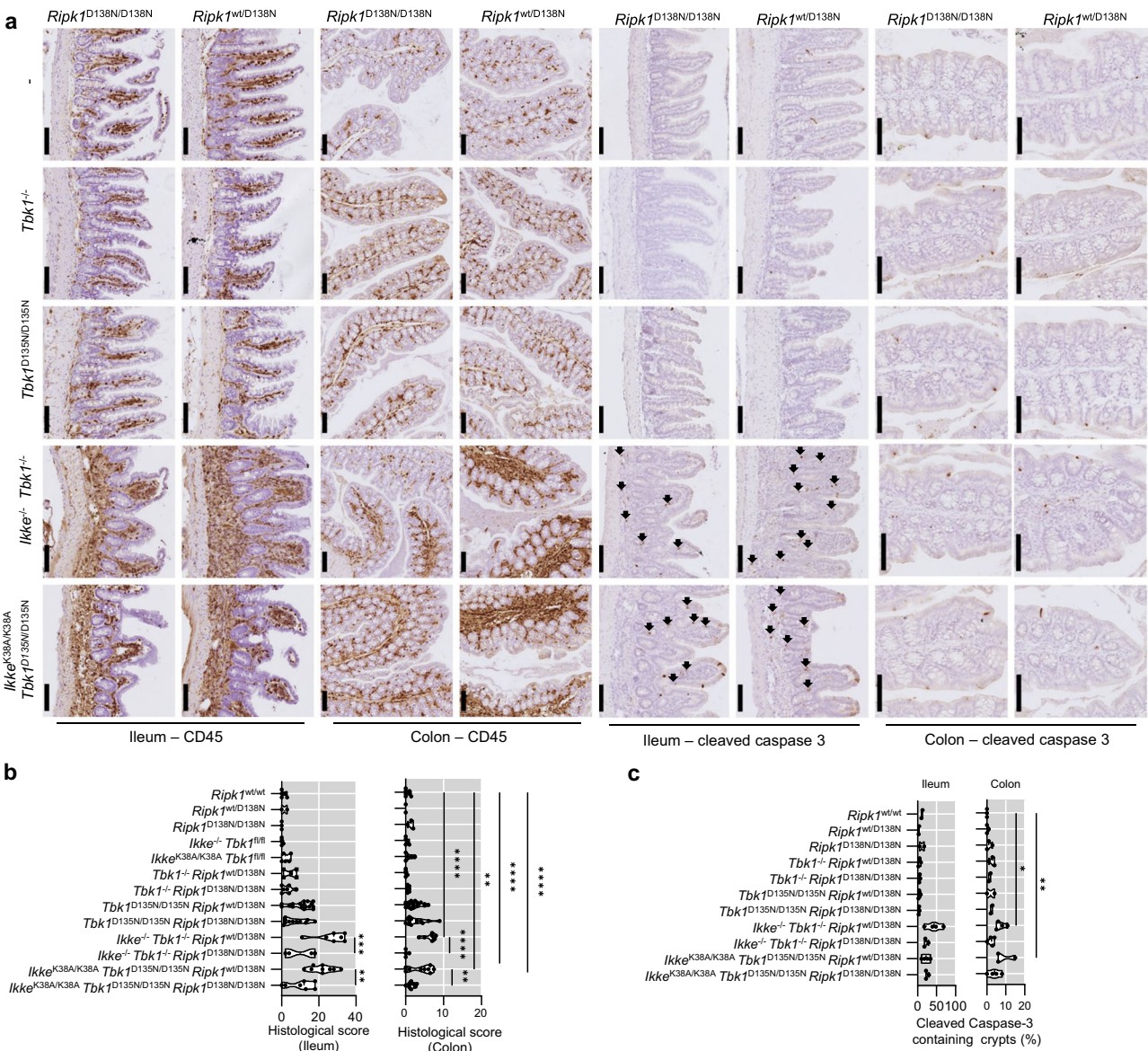

**Fig. 6 | Combined systemic inhibition of TBK1 and IKKε causes intestinal inflammation. a** Representative images of liver sections from 8 to 13 week-old mice with the indicated genotypes immunostained for CD45 and cleaved caspase 3 or stained with hematoxylin and eosin (H&E). Scale bars, 1 mm (black) and 100μm (green). **b, c** Graphs showing histological colitis and ileitis scores **b** and cleaved caspase-3 containing crypt percentage among 100 randomly counted crypts in 8-13 weeks-old mice of the indicated genotypes. Each dot represents one mouse. *p < 0.05, **p < 0.01, ***p < 0.005, ****p < 0.0001. (one-way ANOVA t-test with post-hoc multiple test). Violin plots show the median of biological replicates and interquartile range of data. Source data for **b, c** are provided as a source data file.

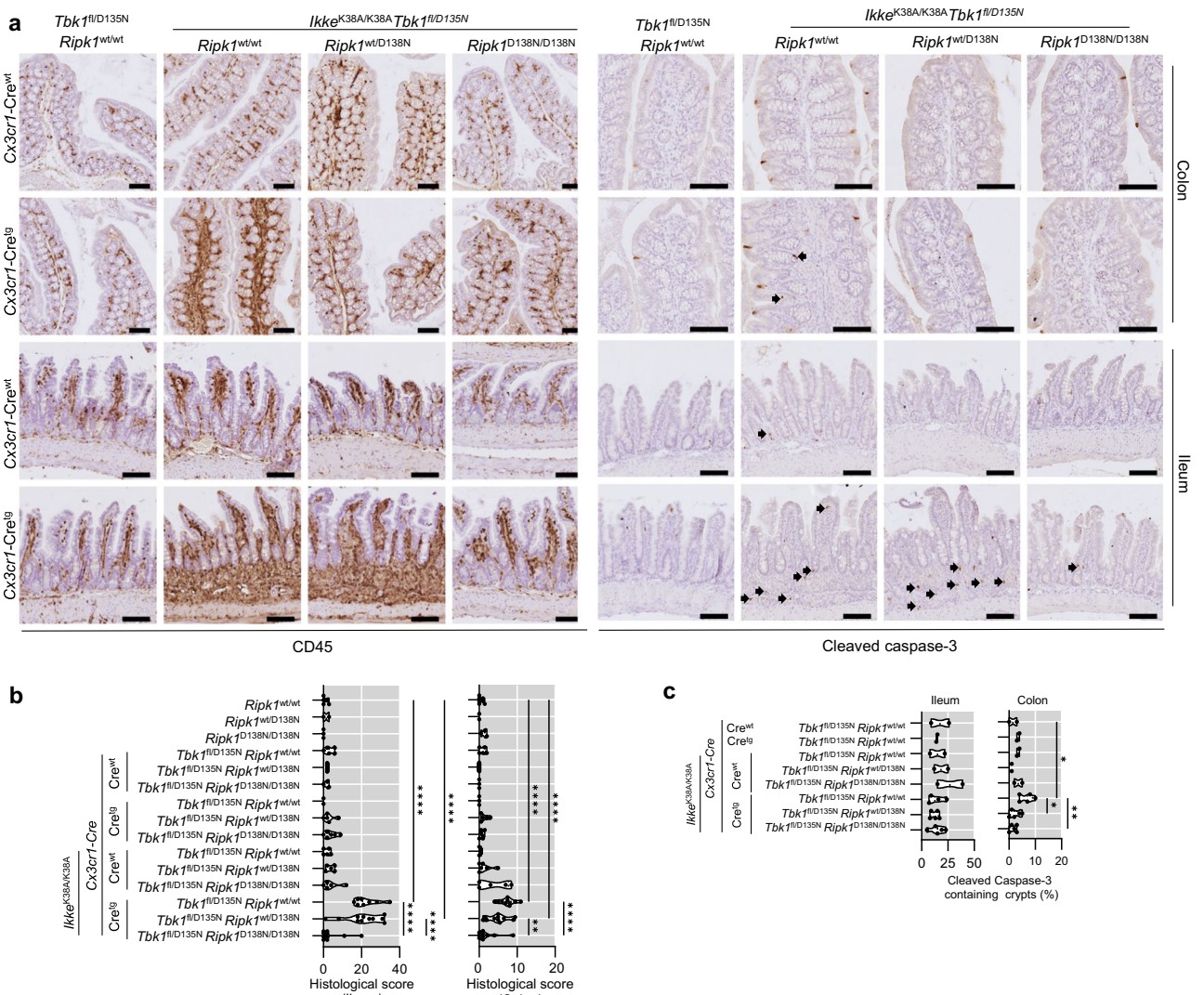

**Fig. 7 | Combined inhibition of TBK1 and IKKε specifically in myeloid cells causes intestinal inflammation. a** Representative images of liver sections from 8-13 week-old mice with the indicated genotypes immunostained for CD45 and cleaved caspase 3 or stained with hematoxylin and eosin (H&E). Scale bars, 1 mm (black) and 100 μm (green). **b, c** Graphs showing histological colitis and ileitis scores **b** and cleaved caspase-3 containing crypt percentage among 100 randomly counted crypts in 8–13 weeks-old mice of the indicated genotypes. Each dot represents one mouse. *$p < 0.05$, **$p < 0.01$, ***$p < 0.005$, ****$p < 0.0001$. (one-way ANOVA t-test with post-hoc multiple test). Violin plots show the median of biological replicates and interquartile range of data. Source data for **b, c** are provided as a source data file.

lesser extent in the colon (Fig. 8a–d and Supplementary Fig. 3a). However, we did not observe pronounced immune cell infiltration in the intestine of these animals (Supplementary Fig. 3b). Importantly, inhibition of RIPK1 kinase activity by homozygous expression of RIPK1-D138N prevented the intestinal pathology of *Ikke*[K38A/K38A] *Tbk1*[fl/D135N] *Villin*-Cre[wt/tg] mice (Fig. 8d). Taken together, these results showed that IKKε and TBK1 act in both IECs and myeloid cells to prevent cell death and inflammation in the gut.

## Inhibition of TBK1 and IKKε causes RIPK1-dependent cell death and IL-1β release in myeloid cells

TBK1 was shown to prevent TNF-induced RIPK1 kinase-dependent cell death in mouse fibroblasts and human cell lines[26,28]. Moreover, genetic TBK1 deficiency in human patients caused systemic inflammation that was ameliorated by treatment with anti-TNF antibodies[27]. Furthermore, TBK1 deficiency in myeloid cells caused elevated pro-IL-1β production in response to TLR-1/2 stimulation with Pam3CSK4[31]. We therefore hypothesized that RIPK1-kinase mediated cell death could

trigger IL-1β production in response to inhibition of TBK1 and IKKε in myeloid cells. To investigate whether TBK1 and IKKε inhibition can lead to cell death and IL-1β release, we treated WT BMDMs with serial dilutions of BX795 and MRT67307, small molecules that inhibit TBK1 and to a lesser extent IKKε kinase activity[32–34]. Indeed, we found that both MRT67307 and BX795 induced cell death and IL1β release from WT BMDMs (Fig. 9a, b). Importantly, MRT67307- and BX795-induced cell death and IL-1β release was prevented by treatment with two specific RIPK1 kinase inhibitors, namely Nec1s and GSK2982772[35,36] (Fig. 9a, b). To genetically confirm the role of RIPK1 kinase activity, we treated WT and *Ripk1*[D138N/D138N] BMDMs with serial dilutions of MRT6707 and measured cell death and IL-1β release. Consistent with the inhibitor studies, we found that *Ripk1*[D138N/D138N] macrophages were protected from cell death and IL-1β release induced by MRT67307 (Fig. 9c, d). MRT67307 and BX795 efficiently inhibit TBK1 but are less potent in suppressing IKKε kinase activity[32–34]. Therefore, to delineate the role of IKKε kinase activity in MRT67307 and BX795-mediated cell death and IL-1β production, we treated WT and *Ikke*[K38A/K38A]

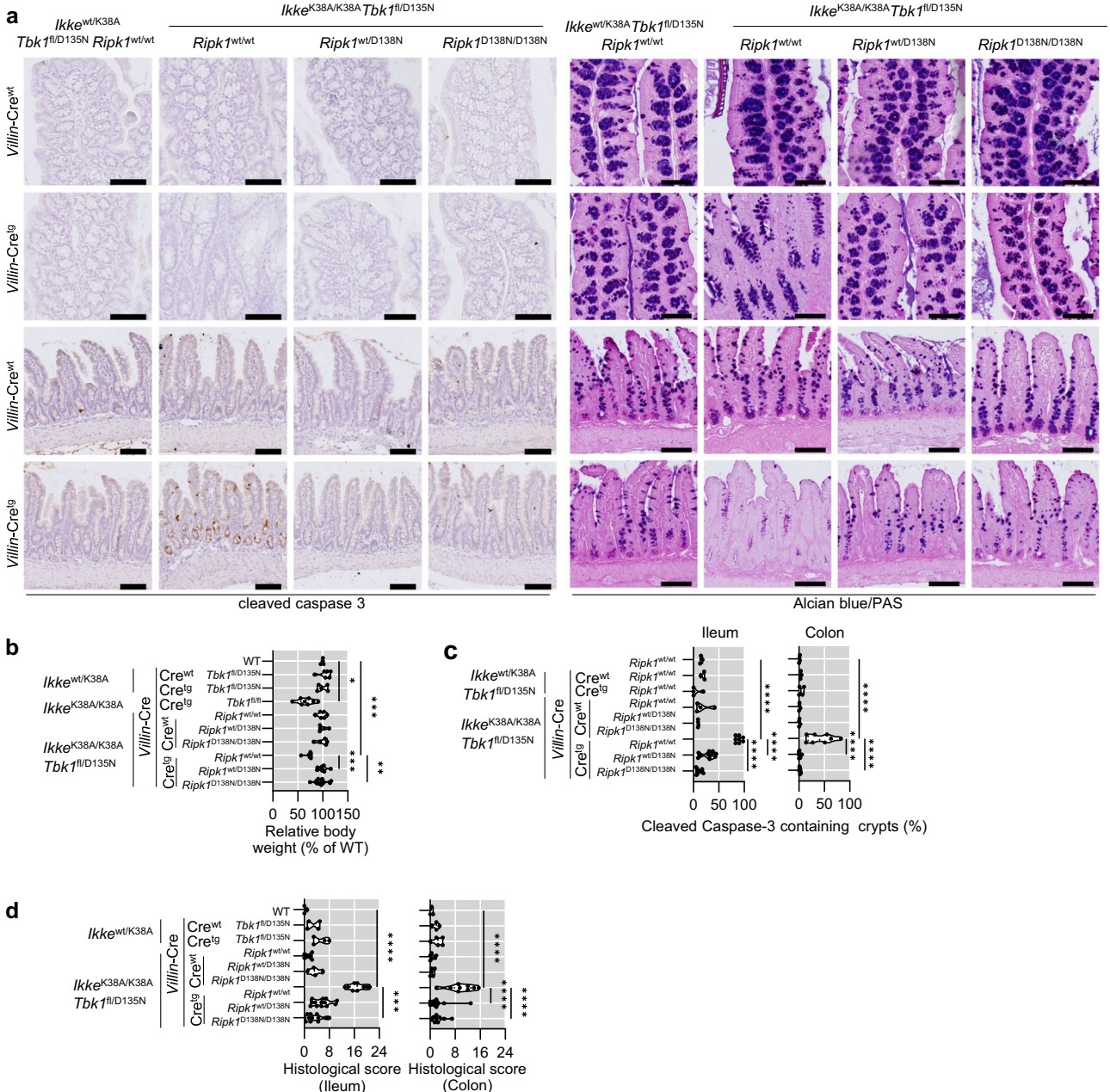

**Fig. 8 | Combined inhibition of TBK1 and IKKε kinase activities specifically in intestinal epithelial cells causes RIPK1-dependent cell death and intestinal inflammation. a** Representative images of colon and small intestine sections of 5–6 week-old mice with the indicated genotypes immunostained with cleaved caspase 3 or stained with Alcian Blue/PAS. Scale bars, 100 μm. **b** Graphs showing relative body weight of 5-6 week-old mice with the indicated genotypes. **c** Graphs displaying the percentage of crypts that have cleaved caspase 3 signal among 100 randomly counted crypts. **d** Graphs showing relative histological colitis and ileitis scores of 5–6 week-old mice with the indicated genotypes. Each dot represents one mouse. $^*p < 0.05$, $^{**}p < 0.01$, $^{***}p < 0.005$, $^{****}p < 0.0001$ (one-way ANOVA t-test with post-hoc multiple test). Violin plots show the median of biological replicates and interquartile range of data. Source data for **b**–**d** are provided as a source data file.

macrophages with serial dilutions of MRT67307 and BX795. These experiments showed that *Ikke*[K38A/K38A] BMDMs showed increased cell death and IL-1β release at lower concentrations of BX795 and MRT67307 compared to WT macrophages (Fig. 9e, f). Taken together, these results showed that inhibition of TBK1 and IKKε triggered RIPK1-dependent cell death and IL-1β release in macrophages.

## IL-1-, IL-18- and IL-33- receptor signaling contributes to the systemic inflammation phenotype of IKKε-kinase dead mice lacking TBK1 kinase activity in myeloid cells

IL-1β but also IL-18 and IL-33 have been shown to promote myelopoiesis and granulocytosis by stimulating G-CSF expression[37–39]. We

therefore hypothesized that increased production of IL-1 family cytokines by TBK1-IKKε-deficient myeloid cells could be implicated in driving the expansion of granulocyte and monocyte compartments and cause systemic inflammation in these mice. Measurement of different cytokines including IL-1 family members in the serum revealed a trend towards increased levels of IL-1β and IL-18 in *Ikke*[K38A/K38A] *Tbk1*[fl/D135N] *Cx3cr1*-Cre[wt/tg] compared to *Ikke*[K38A/K38A] *Tbk1*[fl/D135N] *Cx3cr1*-Cre[wt/tg] *Ripk1*[D138N/D138N] mice (Supplementary Fig. 4a), suggesting that IL-1 family cytokines could be implicated in driving the pathology. To investigate the role of IL-1-, IL-18-, and IL-33- signaling in myeloid cell-driven systemic inflammation in *Ikke*[K38A/K38A] *Tbk1*[fl/D135N] *Cx3cr1*-Cre[wt/tg] mice, we crossed them to mice with combined deficiency of IL-1R1, IL-

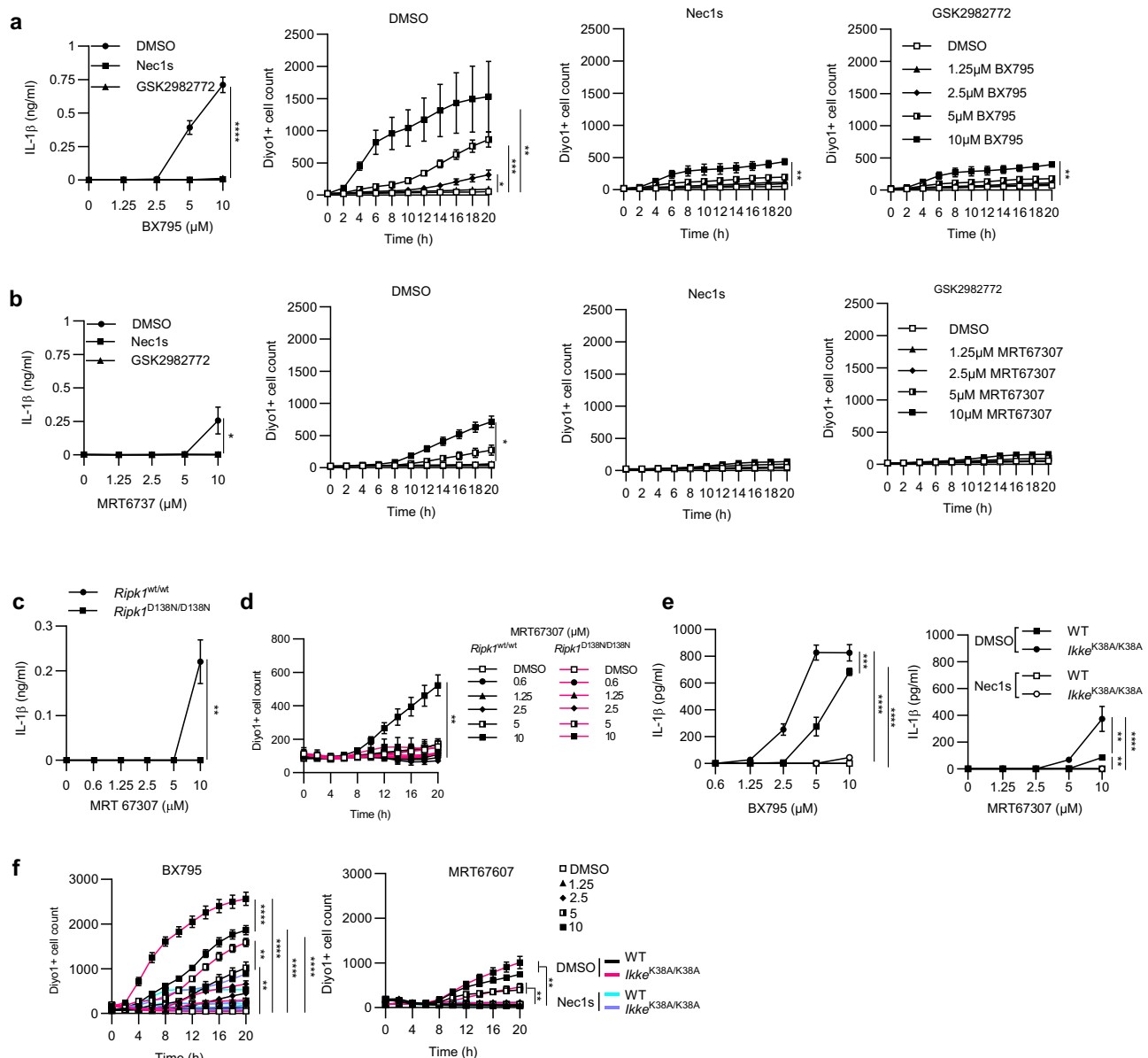

**Fig. 9 | Pharmacological inhibition of TBK1 and IKKε in macrophages causes RIPK1-kinase activity-dependent cell death and IL-1β release. a, b** Graphs showing IL-1β levels in supernatants and Incucyte cell death analysis of BMDMs from wild-type mice stimulated with the indicated amounts of BX795 or MRT67307 in combination with the indicated stimuli, 20μM Necrostatin-s (Nec1s) or 10 μM GSK2982772. IL-1β levels were assessed after 24 **a** or 20 h **b** after stimulation. **c** Graphs showing IL-1β levels in supernatants and Incucyte cell death analysis of BMDMs from mice with the indicated genotypes stimulated with the indicated amounts of MRT67307 for 20 hours. **d** Graphs showing Incucyte cell death analysis of BMDMs from *Ripk1*^wt/wt^ and *Ripk1*^D138N/D138N^ mice treated with indicated

concentrations of MRT67307. **e, f** Graphs showing IL-1β levels in supernatants **e** and Incucyte cell death analysis **f** of BMDMs from mice with the indicated genotypes stimulated for 24 hours with the indicated amount of BX795 and MRT67307 in combination with the indicated stimuli, 20 μM Nec1s or 10 μM GSK2982772. Results shown were obtained from one experiment in which three independent isolations of BMDMs for each genotype (n = 3 biological replicates) were included. Data are the mean of biological replicates with error bars giving SD. *p < 0.05, **p < 0.01, ***p < 0.005, ****p < 0.0001 (two-way ANOVA t-test with post-hoc multiple test). Source data for **a–f** are provided as a source data file.

18R1, and IL-33R (*Il1.18.33r*^-/-^), which were generated using CRISPR/Cas9 gene editing (Supplementary Fig. 4b). Flow cytometry analysis confirmed the absence of IL-18R1 and IL-33R, while immunoblot analysis confirmed the loss of IL-1R1 in BM cells from *Il1.18.33r*^-/-^ mice (Supplementary Fig. 4c, d). In addition, BM cells from *Il1.18.33r*^-/-^ did not produce IL-6 in response to IL-1β, IL-18 and IL-33, but they produced normal levels of IL-6 after stimulation with IL-36β, functionally validating the specific deficiency of IL-1R1, IL-18R1, and IL-33R in these animals (Supplementary Fig. 4e, f).

Analysis of *Ikke*^K38A/K38A^ *Tbk1*^fl/D135N^ *Cx3cr1*-Cre^wt/tg^ *Il1.18.33r*^-/-^ revealed that combined IL-1R1, IL-18R1 and IL-33R deficiency

ameliorated splenomegaly, granulocytosis, and monocytosis, revealing an important role for IL-1, IL-18, and IL-33 in driving systemic inflammation in these animals (Fig. 10a). Interestingly, *Ikke*^K38A/K38A^ *Tbk1*^fl/D135N^ *Cx3cr1*-Cre^wt/tg^ *Il1.18.33r*^-/-^ mice showed a strong decrease in CD69 to CD62L ratio within splenic CD3 + T cells to almost normal levels, demonstrating that IL-1, IL-18, and IL-33 signaling drives T cell activation in response to myeloid-cell specific inhibition of TBK1 and IKKε (Fig. 10a). Combined deficiency of IL-1R1, IL-18R1 and IL-33R also ameliorated inflammation and cell death in the colon but not in the small intestine of *Ikke*^K38A/K38A^ *Tbk1*^fl/D135N^ *Cx3cr1*-Cre^wt/tg^ *Il1.18.33r*^-/-^ mice (Fig. 10b–d). Moreover, the liver of *Ikke*^K38A/K38A^ *Tbk1*^fl/D135N^ *Cx3cr1*-Cre^wt/tg^

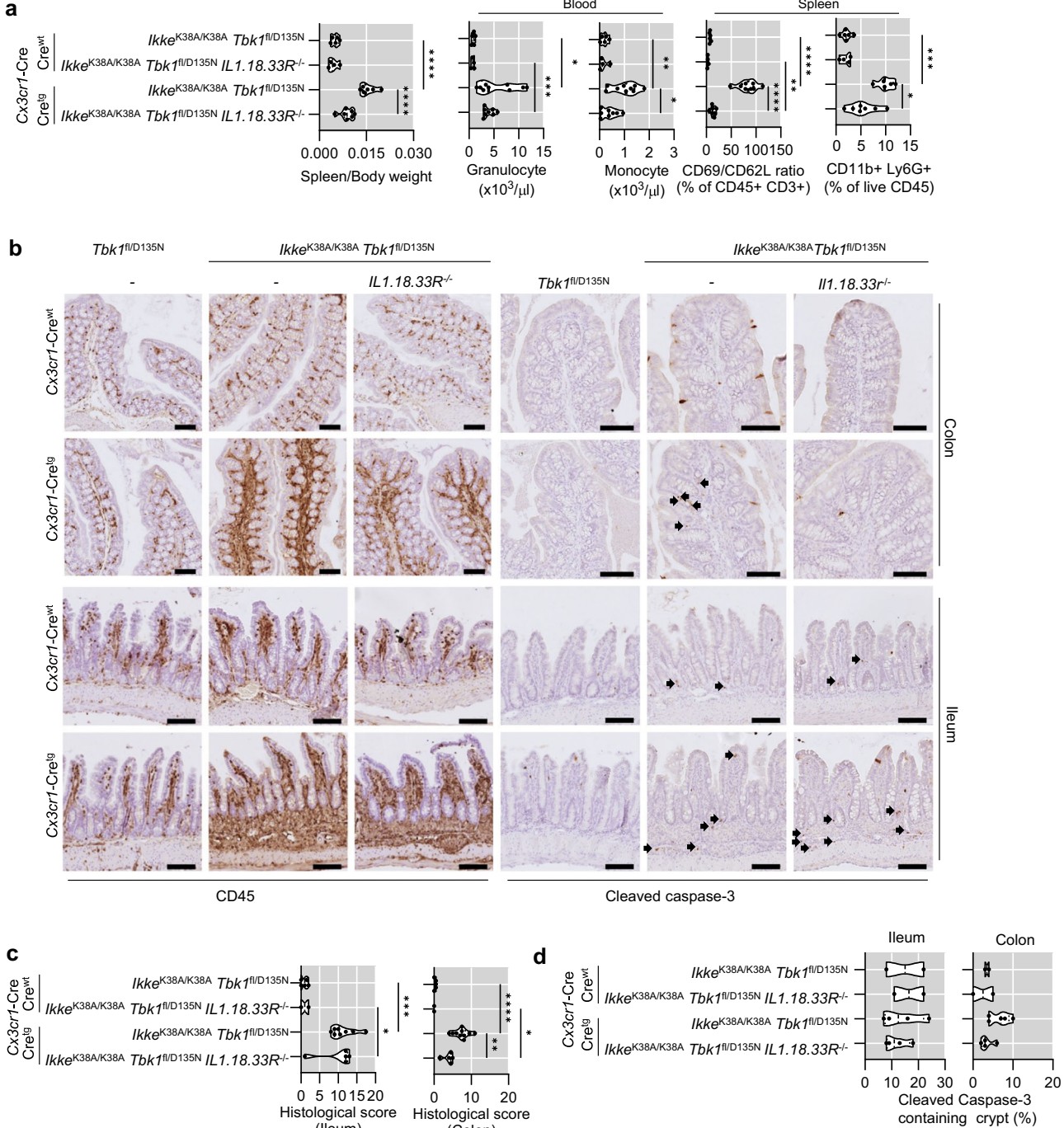

**Fig. 10 | Combined deficiency in IL-1R1, IL-18R1, and IL-33R suppresses systemic inflammation caused by myeloid cell-specific inhibition of TBK1 and IKKε kinase activity. a** Graphs showing spleen to body weight ratio (left panel), peripheral blood analysis assayed by differential hematology analyzer (middle panel), and ratio of CD69+ (activated) to CD62L+ (naïve) T cells amongst live CD45 + CD3+ cells, as well as the percentage of CD11b + Ly6G+ (neutrophils) amongst live CD45+ cells (right panel) in spleens of 8-13 week-old mice with the indicated genotypes. **b** Representative images of ileum and colon sections from 8 to 13 week-old mice with indicated genotypes immunostained for CD45 and cleaved caspase-3. Scale bars, 100 µm. **c** Graphs showing histological colitis and ileitis scores from 8 to 13 week-old mice with indicated genotypes. **d** Graphs displaying the percentage of crypts that have cleaved caspase 3 signal among 100 randomly counted crypts. Each dot represents one mouse. $*p < 0.05$, $**p < 0.01$, $***p < 0.005$, $****p < 0.0001$ (one-way ANOVA t-test with post-hoc multiple test). Violin plots show the median of biological replicates and interquartile range of data. Source data for **a** and **c**, **d** are provided as a source data file.

$Il1.18.33r^{-/-}$ mice showed reduced numbers of cleaved caspase-8 positive cells compared to $Ikke^{K38A/K38A}$ $Tbk1^{fl/D135N}$ $Cx3cr1$-Cre$^{wt/tg}$, however, we did not observe considerable reduction of CD45$^+$ cell infiltration (Fig. 10b). Taken together, these results revealed that IL-1R1, IL-18R1 and IL-33R signaling play an important role in driving myeloid cell expansion and inflammation induced by inhibition of TBK1 and IKKε

kinase activity, however, IL-1R1, IL-18R1, and IL-33R-independent pathways also contribute to the pathology particularly in the ileum.

## Discussion

TBK1 and IKKε are IKK-related kinases that were first identified as critical mediators of type I IFN activation downstream of nucleic acid

sensors[14]. The role of TBK1 in type I IFN activation, metabolic regulation, and TNFR1 signaling has been extensively studied[11,26,27,40–42], however, the role of IKKε remains poorly understood. Here we employed genetic mouse models to study the in vivo role of IKKε and TBK1, particularly focusing on exploring potentially redundant functions of these kinases. Our genetic studies revealed a previously unappreciated function of IKKε in compensating for the loss of TBK1 kinase activity to inhibit RIPK1-mediated cell death, which is important to prevent myeloid cell expansion and multi-organ inflammation. Previous studies employed *Tbk1*[-/-] and *Ikke*[-/-] mice to study the role of these kinases[2,24,25]. However, many kinases, including RIPK1, display both kinase-dependent and kinase-independent functions that are often involved in regulating different cellular functions[16,43]. We therefore generated knock-in mice expressing kinase-inactive mutants of TBK1 (*Tbk1*[D135N/D135N]) and IKKε (*Ikke*[K38A/K38A]) and analyzed these side by side with *Tbk1*[-/-] and *Ikke*[-/-] mice to explore kinase-dependent and -independent functions of these proteins. We found that *Tbk1*[D135N/D135N] mice exhibited embryonic lethality that was rescued by heterozygous or homozygous expression of kinase inactive RIPK1-D138N, similarly to *Tbk1*[-/-] mice as reported previously[26]. *Ikke*[K38A/K38A] mice were born normally and did not develop pathology, consistent with previous studies on *Ikke*[-/-] mice[2]. The combined loss of TBK1 and IKKε (*Tbk1*[-/-] *Ikke*[-/-]) or their kinase activities (*Tbk1*[D135N/D135N] *Ikke*[K38A/K38A]) also caused embryonic lethality that was rescued by heterozygous or homozygous expression of kinase inactive RIPK1-D138N. This finding is surprising in light of a previous report that TNF deficiency could rescue the embryonic lethality of *Tbk1*[-/-] mice but not of double deficient *Tbk1*[-/-] *Ikke*[-/-] mice[25], arguing that combined loss of both IKK-related kinases caused death during development that is at least in part mediated by TNF-independent mechanisms. Our results suggest that TBK1 and IKKε act in a functionally redundant fashion during development to prevent RIPK1 activation downstream of multiple receptors in addition to TNFR1. Inhibition of RIPK1 kinase activity was shown to block necroptosis downstream of Fas, TRAIL-R as well as TRIF-mediated TLR3/4 signaling[18,44,45], suggesting that these pathways may be implicated in causing embryonic lethality upon combined loss of TBK1 and IKKε. Interestingly, a recent study reported that TBK1 deficiency did not cause embryonic lethality but resulted in TNF-mediated autoinflammatory pathology in human patients, suggesting that the role of IKKε and TBK1 during embryonic development may be different in mice and humans, although these findings also need to take into account potential effects of genetic heterogeneity in the human population compared to inbred mouse strains[27].

*Tbk1*[-/-] *Ripk1*[wt/D138N] or *Tbk1*[D135N/D135N] *Ripk1*[wt/D138N] mice reached adulthood and did not develop apparent pathology at least up to the age of 4-5 months, showing that inhibition of RIPK1 kinase activity by about 50% was sufficient to fully inhibit cell death and inflammation induced by loss of TBK1 also in adult mice. In contrast, *Tbk1*[D135N/D135N] *Ikke*[K38A/K38A] *Ripk1*[wt/D138N] as well as *Tbk1*[-/-] *Ikke*[-/-] *Ripk1*[wt/D138N] mice showed a number of pathological features affecting multiple tissues, revealing an important role of IKKε in compensating for the loss of TBK1 to maintain tissue homeostasis and prevent disease. Combined loss of TBK1 and IKKε or their kinase activities in *Ripk1*[wt/D138N] genetic background caused transient alopecia, with 100% of these animals showing lack of hair at 3-4 weeks of age but fully recovering hair growth by the age of 7-8 weeks. The transient alopecia induced by combined loss of TBK1 and IKKε was entirely dependent on RIPK1 kinase activity as it was not observed in a homozygous *Ripk1*[D138N/D138N] genetic background. Interestingly, *Ikke*[K38A/K38A] *Tbk1*[fl/D135N] *K14-Cre*[Tg/wt] mice did not show hair loss arguing that TBK1 and IKKε act in cells other than keratinocytes to prevent alopecia. It is also noteworthy that keratinocyte-specific inhibition of TBK1 and IKKε did not cause skin pathology even in the presence of wild type RIPK1, showing that IKK-related kinases are not required to prevent RIPK1-mediated cell death and inflammation in the skin. We showed previously that keratinocyte-specific IKK2 knockout

caused severe skin inflammation mediated by TNFR1-induced, RIPK1-dependent cell death, suggesting that IKK2 is the main kinase responsible for suppressing RIPK1 activation in keratinocytes[46–48].

*Tbk1*[D135N/D135N] *Ikke*[K38A/K38A] *Ripk1*[wt/D138N] and *Tbk1*[-/-] *Ikke*[-/-] *Ripk1*[wt/D138N] mice showed splenomegaly and a pronounced expansion of monocytes and neutrophils in peripheral blood and lymphoid tissues, which was largely dependent on RIPK1 kinase activity as this phenotype was strongly, but not completely, suppressed in a *Ripk1*[D138N/D138N] background. Considering that *Tbk1*[D135N/D135N] *Ripk1*[wt/D138N] or *Tbk1*[-/-] *Ripk1*[wt/D138N] mice did not show hematological abnormalities, these results revealed an important role of IKKε in compensating for the loss of TBK1 to prevent expansion of myeloid cells via RIPK1-dependent and -independent mechanisms. Myeloid cell-specific inhibition of TBK1 and IKKε caused splenomegaly, neutrophilia, and monocytosis that were dependent on RIPK1 kinase activity, demonstrating that TBK1 and IKKε function in a cell-intrinsic manner to suppress RIPK1-dependent expansion of myeloid cells. We found that inhibition of TBK1 and IKKε induced RIPK1-dependent cell death and IL-1β release in BMDMs, suggesting that IL-1β, which is known to promote myeloid differentiation, could be implicated in causing neutrophilia and monocytosis. Indeed, combined deficiency of IL-1R1, IL-18R1, and IL-33R strongly suppressed splenomegaly and myeloid cell expansion induced by myeloid cell specific inhibition of TBK1 and IKKε, providing evidence that RIPK1-mediated cell death resulting in IL-1 family cytokine production by myeloid cells contributes to the pathology.

*Tbk1*[D135N/D135N] *Ikke*[K38A/K38A] *Ripk1*[wt/D138N] and *Tbk1*[-/-] *Ikke*[-/-] *Ripk1*[wt/D138N] mice developed liver pathology, characterized by elevated markers of liver damage in the serum concomitant with increased cell death and infiltration of immune cells in the tissue. Inhibition of RIPK1 kinase activity by a *Ripk1*[D138N/D138N] genetic background strongly suppressed cell death and immune cell infiltration, showing that combined systemic inhibition of TBK1 and IKKε caused liver pathology mediated mainly by RIPK1 kinase activity. Surprisingly, liver parenchymal cell-specific inhibition of TBK1 and IKKε kinase activities did not cause cell death and immune cell infiltration in the liver. Notably, liver parenchymal cell specific IKK2 deficiency also did not induce spontaneous liver pathology in contrast to NEMO knockout in LPCs that caused hepatocyte RIPK1-mediated apoptosis, chronic liver damage, and the development of hepatocellular carcinoma[49–51]. Together, these findings suggest that NEMO may act as an adapter engaging both TBK1/IKKε and IKK2, which act in a functionally redundant fashion to suppress RIPK1-dependent hepatocyte apoptosis and liver damage. Interestingly, mice with myeloid-cell specific inhibition of TBK1 and IKKε showed increased cell death and immune cell infiltration in the liver, although these animals did not have elevated liver damage markers in the serum. Therefore, only systemic inhibition of TBK1 and IKKε could cause clinically relevant liver damage, arguing that loss of function in multiple cell types is required to cause the pathology. It is possible that inhibition of TBK1 and IKKε in myeloid cells, which induces immune cell infiltration in the liver, may synergize with loss of TBK1 and IKKε function in liver parenchymal cells that sensitizes hepatocytes to TNF-induced cell death, to aggravate the liver pathology.

Combined systemic inhibition of TBK1 and IKKε kinase activity also caused intestinal pathology, manifesting with increased cell death and immune cell infiltration in both the ileum and colon of *Tbk1*[D135N/D135N] *Ikke*[K38A/K38A] *Ripk1*[wt/D138N] and *Tbk1*[-/-] *Ikke*[-/-] *Ripk1*[wt/D138N] mice. Importantly, homozygous expression of RIPK1D138N could prevent cell death and immune infiltration in the colon but only partially ameliorated the pathology in the ileum. These findings showed that IKKε has an important role in compensating for the loss of TBK1 to prevent intestinal inflammation by inhibiting both RIPK1-dependent and RIPK1-independent mechanisms. Our genetic studies revealed that TBK1 and IKKε act in both myeloid cells and intestinal epithelial cells to prevent intestinal inflammation. Mice with myeloid cell specific TBK1

and IKKε inhibition showed ileitis and colitis that were almost completely inhibited by homozygous expression of kinase-inactive RIPK1. Furthermore, mice with intestinal epithelial cell-specific inhibition of TBK1 and IKKε also developed intestinal pathology characterized by death of IECs, loss of Paneth cells, reduced numbers of Goblet cells and increased immune cell infiltration in both the ileum and colon, which were prevented by crossing to *Ripk1*[D138N/D138N] genetic background. Taken together, these results showed that TBK1 and IKKε act in a redundant fashion in both myeloid and intestinal epithelial cells to prevent RIPK1-dependent inflammation in the intestine. Notably, IEC-specific ablation of NEMO, but not of IKK2, induced spontaneous intestinal inflammation by sensitizing IECs to RIPK1-dependent cell death[52,53]. These findings suggest that TBK1 and IKKε have a more important role than IKK2 in preventing RIPK1-mediated death of IECs, although the intestinal phenotype of *Nemo*[fl/fl] *Villin*-Cre[wt/tg] mice is more severe than that observed in *IKKε*[K38A/K38A] *Tbk1*[fl//D135N] *Villin*-Cre[wt/tg], indicating that NEMO likely engages both TBK1/IKKε and IKK2 to protect IECs from TNFR1-induced cell death.

Taken together, our genetic studies revealed an important role of IKKε in compensating for the absence of TBK1 to prevent cell death and inflammation in multiple tissues. Importantly, RIPK1 kinase activity inhibition strongly suppressed, but could not fully prevent, the pathologies induced by combined inhibition of TBK1 and IKKε, showing that TBK1/IKKε also inhibit cell death and inflammation driven by RIPK1 kinase-independent mechanisms. In addition to its kinase activity-dependent cell death-inducing functions, RIPK1 also acts as a scaffold to promote proinflammatory signaling, which could be implicated in driving inflammation in mice lacking TBK1 and IKKε kinase activities. Moreover, TBK1 and IKKε are implicated in the regulation of multiple inflammatory signaling pathways[2-5,7-10,54], metabolism[42], mitochondrial homeostasis[41,55,56] and autophagy[57]. While these functions of IKKε and TBK1 could be involved in driving the RIPK1 kinase-independent pathology caused by combined inhibition of TBK1 and IKKε, the specific underlying mechanisms remain elusive and remain to be elucidated in future studies. Our results also provide important insights into the cell-specific function of these kinases, that could also be relevant for the better understanding of the mechanisms driving the autoinflammatory pathology developing in humans with mutations disrupting TBK1 expression[27]. TBK1 and IKKε have an important role in myeloid cells and intestinal epithelial cells, where they act to prevent RIPK1-mediated cell death and inflammation, but are dispensable in liver parenchymal cells and epidermal keratinocytes. Together with earlier studies of the tissue-specific function of NEMO and IKK2[46-53,58], these findings suggest that NEMO suppresses RIPK1-mediated cell death and inflammation by recruiting both canonical and non-canonical IKKs, which exhibit partly overlapping but also tissue-specific functions.

## Methods
### Mice
The generation of Ripk1[D138N/D138N 18], *Tbk1*[fl/fl 29], CMV-Cre[30], *Vil1*-Cre[59], *Cx3cr1*-Cre[60], *Afp*-Cre[61], and *Keratin14*-Cre[62], mice were previously described. *Tbk1*[WT/D135N], *Ikke*[K38A/K38A], *Ikke*[-/-], and *Il1.18.33r*[-/-] mice were generated using CRISPR/Cas9-mediated gene targeting. For *Tbk1*[D135N] mice, a short-guide (sg)RNA (5′- TGT TGC CTG GCT TGA TAT CT -3′) and long single stranded(ss) oligonucleotide (5′- AAA TCC GTG AGT TTG TAC ACA GAC TGG CCG TCC TCC CCT ATG ACG CGC ATG ATG TTG CCT GGC TTG ATA TtT CGa TGC ACT ATG CCG TTC TCT CGG AGA TGA TTC ATC CCG CCC ACT TCA GCA ATA GGT ACA AAA CAA GG -3′); for *Ikke*[K38A/K38A] and *Ikke*[-/-] mice a sgRNA (5′- CGGGGAGGTGGTTGC TGTAA-3′) and long ss oligonucleotide (5′- GCA GAG GTG GTC CCC ATT GCT TAC TCT CGC CCT GTG CCC ACG CTA GAA ATC CGG GGA GGT GGT TGC TGT AGc tGT CTT CAA CTC AGC CAG CTA TCG GCG ACC TCC TGA GGT TCA GGT GAG GGA GTT TGA GGT CCT GCG GAG GC -3′); for

*Il1.18.33r*[-/-] mice sgRNAs (5′- AGA CCC CCA TAT CAG CGG AC -3′ targeting *IL-1R1*, 5′- GCC ACC ATG AGA TGG TTC AA -3′ targeting *IL-18R1*; 5′- AGG ACG CTC GAC TTA TCC TG -3′ targeting *IL-33R*; and 5′- ATA CCC GCC AGA AAC AAA CG -3′ targeting *IL-36R*) were co-injected into fertilized wild-type oocytes together with Cas9 protein and mRNA.

For genotyping of *Ikbke*[K38A] mice, a 330 bp region spanning the mutation site of *Ikbke* was amplified using Universal Forward (5′- ACC CTT GAG GGA CAT CAG GT -3′) and Universal Reverse (5′- CGA TGT TCT GGT GAT TCA GCC T -3′), and WT (259 bp) and mutant (107 bp) sequence were respectively amplified using WT Reverse (5′- GCT GGC TGA GTT G AA GAC CTT T -3′) and (5′- GGA GGT GGT TGC TGT AGC TG -3′) primers in a single PCR reaction. TBK1-D135N mutation destroyed the EcoRV restriction site on this genomic region. For genotyping of TBK1[D135N], the amplicon generated with primers (5′- ACT CCA CAT AGA ACG TGC CTC -3′) and (5′ ACA CAC TTG TGC CTG AGG ATT -3′) for TBK1[D135N] and incubated with EcoRV-HF (Neb, cat no R3195) restriction enzyme for 90 minutes at 37°C. For genotyping of *Il1.18.33r*[-/-] mice, primer mixture consists of (5′- AGC CAC CAT GAG ATG GTT CAA -3′) and (5′- TGC ATG GAT TAT CAT GAG CTT -3′) primers or (5′- AAA CAG CCA CCA TGA TTG TGA G -3′) and (5′- TGCATGGATTATCATGAGCTT -3′) primers were used to amplify WT and mutant allele, respectively. PCR amplicons were visualized on 2.0% agarose gels stained with ethidium bromide.

Mice were maintained under specific-pathogen-free (SPF) conditions in the animal facilities of the Institute for Genetics and the CECAD Research Center of the University of Cologne. All mouse procedures were conducted in accordance with national and institutional guidelines, and protocols were approved by the responsible local authorities in Germany (Landesamt für Natur, Umwelt und Verbraucherschutz Nordrhein-Westfalen). Mice of the indicated genotype were randomly assigned to groups, and experiments were not blinded. Both male and female mice are included in all groups.

### Blood analysis
Blood-cell-count analysis was performed using 40 µl of peripheral blood samples from mice using an Abacus Junior Vet analyzer according to the manufacturer's instructions.

### Isolation and culture of bone marrow-derived macrophages
For the preparation of BMDMs, bone marrow cells were plated on 145 mm bacterial Petri dishes (Greiner, cat no 639161_120) in DMEM medium (Invitrogen, cat no 41965062) supplemented with 20 ng/ml mM-CSF (ImmunoTools, cat no. 12343113), 10% fetal calf serum, penicillin/streptomycin, and sodium pyruvate. The cells were plated in tissue-coated dishes on Day 6 of the culture, and the experiments were performed on Day 7.

### Cell culture
BMDMs were stimulated with MRT67307 (Selleckchem, cat no. S7948), BX795 (Medchemexpress, cat no. HY-10514), 10 µM GSK2982772 (Selleckchem, cat no. S8484), or 20µM Necrostatin-s1 (Selleckchem, cat no. S8641) for the indicated times in the presence of 20 ng/ml mM-CSF.

### Cell death assays
BMDMs were seeded overnight prior to stimulation in a 96-well plate (Sigma-Aldrich, cat no. CLS3340 or TPP cat no. TPP92096) (5 × 10⁴ cells per well). On the day of the experiment, indicated stimuli were added in the presence of 0.25 µM Diyo-1™ (ATT Bioquest, cat no 17580). Cell death assay was performed using the Incucyte bioimaging platform (Essen); four images per well were captured with a lens that has a magnification of 10. Cell death was measured by Diyo-1 incorporation using Incucyte software.

## Cytokine analysis

The cytokine levels in the supernatants from macrophage cultures were determined by ELISA kits for IL-1β (Thermo Scientific, cat. no. 88–7013–88) and IL-6 (Thermo Scientific, cat no. 88-7064-88) according to the manufacturer's instructions. The serum levels of IL-1α, IL-1β, IL-18, IL-33, IL-6 and IP-10 were measured using ProcartaPlex multiplex assay (Thermo Scientific) according to the manufacturer's instructions

## Immunoblotting

Cell lysates were denatured in 2×Laemmli buffer (Biorad, cat no 34095) supplement with 4% β-mercaptoethanol and 10 mM Dithiothreitol. The protein samples were subsequently boiled at 95 °C for 4 minutes and separated by NuPage™ (Thermo Scientific, cat. no. NP0336BOX) or Tris-polyacrylamide gel. Separated proteins were transferred to PVDF membranes and were blocked using SuperBlock™ blocking buffer (Thermo Scientific, cat no.37515). Incubation of the membranes with primary antibodies was performed in TBS supplemented with 0.1% Tween-20 (v/v), 5% (w/v) BSA and 0.01% sodium azide. Incubation with secondary antibody and washing of the membrane was done in TBS supplemented with 0.1% Tween-20 (v/v) and 5% (w/v) non-fat dry milk. For re-blotting, the membranes were stripped using Restore™ Western Blot Stripping Buffer (Thermo Scientific, cat. no. 21059). The immunoblots were incubated overnight with primary antibodies against p-IRF3 Ser396 (Thermo Scientific, cat no MA5-14947, 1:2000), IRF3 (CST, cat no. 4302, 1:2000), IKKε (CST, cat no. 3416, 1:2000), TBK1 (CST, cat. no. 3013, 1:2000), IL-1R1 (Santa Cruz, cat no sc-393998, 1:2000), GAPDH (Novus biologicals, cat. No NB300,1:10000), vinculin (CST, cat. no. 13901,1:10000), alpha-tubulin (Sigma Aldrich, cat. no. T6074, 1:10000), anti-mouse secondary antibody (Amersham Pharmacia, cat. no. NA931V, 1:5000) and anti-rabbit secondary antibody (Amersham Pharmacia, cat. No NA934V, 1:5000) was used to detect protein using chemiluminescence (Thermo Scientific, cat no 34578 or Thermo Scientific cat no. 34095). Chemiluminescent signal on membranes was measured using Vilber FX system.

## Biochemical serum analysis

Serum alanine transaminase (ALT), alkaline phosphatase (ALP), and aspartate transaminase (AST) were measured in blood serum using a standard assay in a Cobas C111 biochemical analyzer (Roche, Mannheim, Germany).

## Tissue preparation

The colon and small intestine were cut open longitudinally, and the lumen was washed with PBS. The colon tissue and ~8-9 cm of the distal small intestine, designated as the ileum, tissues were Swiss-rolled and fixed in 4% PFA overnight.

## Immunohistochemistry and histology

Samples from mice were fixed in 4% PFA and embedded in paraffin. Sections of 5 μm were subjected to histological analysis by H&E staining or Alcian Blue and PAS staining (Abcam, cat no. ab245876) according to the manufacturer's instructions. For immunohistochemical analysis, slides were rehydrated and incubated with peroxidase blocking buffer (0.04 M Sodium Citrate, 0.121 M $Na_2HPO_2$, 0.03 M $NaN_3$, 3% $H_2O_2$) for 15 minutes. Slides were washed, and antigen-retrieval was performed by digestion with proteinase K (10 μg/ml) for 5 min in TEX-buffer (50 mM Tris, 1 mM EDTA, 0.5 % Triton X-100, pH 8.0) for CD45 and lysosome or with Sodium citrate buffer in a pressure cooker for 20 minutes before the IHC staining. Sections were blocked for 60 minutes using commercial blocking buffer (ThermoFisher, cat no.37536) and incubated with the primary antibody CD45 (eBioscience, cat. no. 14-0451, 1:500), cleaved caspase-3 (CST, cat. no. 9661, 1:500), cleaved caspase-8 (CST, cat no.

8592, 1:500), or lysosome (Dako, cat no. F0372, 1:500) overnight. A biotinylated secondary anti-rat antibody (Jackson Immuno Research, 112-065-003, 1:500) or anti-rabbit (Molecular Probe, cat no. B2770, 1:500) was incubated in the sections for 60 minutes. Stainings were visualized using the ABC Vectastain Elite Kit (Vector Laboratories) and DAB substrate (Dako and Vector Laboratories). Sections were then counterstained with hematoxylin for staining of the nuclei, dehydrated, and mounted with Entellan.

## Histopathological analysis

The histopathological score was evaluated using 3 μm thick H&E-stained sections of paraffin-embedded Swiss-rolled intestinal tissues, using a modified version of the scoring system described in (Adolph et al., 2013). It was explained in detail in our previous publication[63]. Briefly, histopathology scores consist of four parameters: epithelial hyperplasia, quantity, and localization of tissue inflammation, epithelial cell death, and epithelial injury. An "area factor" for a fraction of affected tissue was considered and multiplied with the respective parameter score. Area factor is 1, 2, 3, and 4 for 0–25%, 25–50%, 50–75%, and 75–100% of the affected tissue in terms of aforementioned parameters, respectively. The histology score is the sum of all their area factors with corresponding parameter scores, and the highest score can be 64. Histological scores were based on one Swiss-roll section per mouse and were performed in a blinded fashion.

## Flow Cytometry

The entire spleen or entire thymus was homogenized using 0.40 μm filter and syringe plunger in the presence of Flow medium (1xPBS and % 4 FCS). Bone marrow was extracted from one leg, a femur, and a tibia. Red blood cells were lysed using homemade red blood cell lysis pH7.2 buffer (9:1 8.3 g/l NH4CL and 0.17 M Tris) or ACK cell lysis buffer (Thermoscientific, cat. no. A1049201). $2×10^6$ cells were stained with different set of fluorochrome conjugated antibodies: (i) anti-CD4 (eBioscience, cat no. 25-0041-81, 1:200), anti-CD8 (eBioscience, cat no. 11-0081-82, 1:100), anti-CD3e (eBioscience, cat no. 45-0031-82, 1:200), anti-CD45 (eBioscience, cat. no. 67-0451-82, 1:200), anti-CD62L (eBioscience, cat. no. 17-0621-82, 1:200), anti-CD69 (eBioscience, cat no. 12-0691-82, 1:200); (ii) anti-CD45 (eBioscience, cat. no. 67-0451-82, 1:200), anti-Ly6G (eBioscience, cat no. 46-9668-82, 1:100), anti-CD19 (eBioscience, cat no. 78-0193-82, 1:200), anti-CD11b (BD Horizon, cat no. 562317, 1:100); (iii) anti-CD19 (eBioscience, cat no. 63-5941-82, 1:100), anti-CD3e (eBioscience, cat no. 63-0031-82, 1:100), anti-B220 (eBioscience, cat no. 63-0452-82, 1:100), anti-NK1.1(eBioscience, cat no. 63-5941-82, 1:100), anti-CD115 (BioLegend, cat no. 53-1152-82, 1:50), anti-Ly6G cat no. 46-9668-82, 1:100), anti-CD11b (BD Horizon, cat no. 562317, 1:100); (iv) anti-CD3 (Invitrogen, cat. no. HM3428, 1:100), anti-Foxp3 (Thermoscientific, cat. no. 12-5773-82, 1:20), anti-CD25 (BioLegend, cat. no. 25-0251-82, 1:100), anti-CD4 (BioLegend, cat. no. 17-0042-82, 1:100); (v) anti-CD4 (eBioscience, cat. no. 25-0041-81, 1:100), anti-CD8 (eBioscience, cat. no. 11-0081-82, 1:200), anti-CD3e (eBioscience, cat no. 45-0031-82, 1:200), anti-CD45 (eBioscience, cat. no. 67-0451-82, 1:200); and (vi) anti-CD45 (eBioscience, cat. no. 67-0451-82, 1:200) with anti-IL18Ra (Thermoscientific, cat. no. 157903, 1:100) or anti-IL-33R (Biolegend, cat no. 145303, 1:100) in the presence of Live/Dead fixable yellow dead cell stain (Invitrogen, cat no. L34959) and CD16/32 (BD, cat no. 553142). Stained cells were acquired in SR Fortessa (BD Biosciences), and data were analyzed using FACSDiva software (BD Biosciences) or FlowJo software.

## Quantification and Statistical analysis

Statistical analysis was performed with GraphPad Prism. Significance were determined by one-way ANOVA test or two-way ANOVA test with post hoc multiple comparison test. $p < 0.05$, **$p < 0.01$, ***$p < 0.005$, ****$p < 0.0001$ for all figures. Violin plots show the median of biological

replicates and interquartile range of data, and data in the rest of graphs are the mean of biological replicates with the error bars giving SD. No statistical method was used to predetermine sample size. No data were excluded from the analyses.

### Reporting summary

Further information on research design is available in the Nature Portfolio Reporting Summary linked to this article.

## Data availability

The datasets generated during and/or analyzed during the current study are available from the corresponding author upon reasonable request. Source data are provided in this paper.

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

## Acknowledgements

We thank J. Kuth, C. Uthoff-Hachenberg, E. Stade, E. Gareus, P. Roggan, L. Elles, M. Hahn, and J. von Rhein for technical assistance. We also thank B. Zevnik and the CECAD Transgenic Core Facility for CRISPR/Cas9-assisted gene targeting in mice. Research reported in this publication was supported by funding from the European Research Council (ERC) under the European Union's Horizon 2020 research and innovation programme (Grant Agreement No. 787826), and the Deutsche Forschungsgemeinschaft (DFG, German Research Foundation, projects SFB1403 (Project No. 414786233) and SFB1399 (Project No. 413326622), and under Germany's Excellence Strategy–EXC 2030 CECAD (project no. 390661388) to M.P. R.O.E was supported by a postdoctoral fellowship from the Alexander von Humboldt foundation.

## Author contributions

Conceptualization R.O.E. and M.P.; Methodology R.O.E. and M.P.; Investigation R.O.E., G.G.K., R.S., and M.P.; Writing – Original Draft R.O.E. and M.P.; Writing – Review & Editing R.O.E. and M.P.; Funding acquisition R.O.E. and M.P.; Formal analysis R.O.E. and G.G.K.; Visualization R.O.E. and G.G.K.

## Funding

## Competing interests

R.O.E. and M.P. are co-inventors on a patent application filed by the University of Cologne that is related to this work. The other authors declare no competing interests.
