## [Peer Review File · Nature Communications]

IKKe and TBK1 prevent RIPK1-dependent and -independent inflammationREVIEWER COMMENTS

Reviewer #1 (Remarks to the Author):

This is an elegant study that builds on previous observations from humans and mice that demonstrate that TBK1 has a critical role in preventing TNFR1-mediated and RIPK1-dependent cell death. The authors generated mice with the combined loss of IKK ϵ and TBK1 which causes embryonic lethality and evaluate the role of RIPK1 dependent and independent mechanisms in regulating embryonic development and tissue homeostasis.

RIPK1 kinase activity inhibition strongly suppressed, but could not fully prevent, the pathologies induced by combined inhibition of TBK1 and IKK ϵ , showing that TBK1/IKK ϵ also inhibit cell death and inflammation driven by RIPK1-independent mechanisms. In addition, the authors also reveal unique cell-type specific function of these kinases. TBK1 and IKK ϵ have an important role in myeloid cells and intestinal epithelial cells, where they limit RIPK1-mediated cell death and inflammation, but are dispensable in liver parenchymal cells and keratinocytes. Overall this is an extensive and comprehensive rigorous evaluation of combined loss of these two IKK related kinases and the studies are very clear and compelling.

Reviewer #2 (Remarks to the Author):

In the article by Eren et al., the authors use several genetic crosses to examine the functional redundancies and relative roles of the related kinases TBK1 and IKK ϵ in cellular homeostasis. As the authors note, the cross-regulation between TBK1 and RIPK1 has been extensively studied to characterize the role of TBK1 in TNFR and IFN signaling, and the authors instead focus on characterizing TBK1's redundancies with the poorly studied IKK ϵ kinase in regulating the homeostasis of RIPK1 signaling in development, cell death, and inflammation. The authors also show a key role for the IL-1 family of cytokines in driving the RIPK1-mediated pathology, suggesting potential therapeutic targets for RIPK1-dependent pathologies. While the manuscript is primarily observational, the phenotypes are of interest, and the findings could be improved by additional mechanistic characterization.

Major comments

1. Given the critical role for TBK1 in IFN signaling to drive auto-inflammatory diseases and the spontaneous development of RIPK1-dependent disease in the context of caspase-8 deficiency (PMID: 29666472, PMID: 31827281, PMID: 31511692, PMID: 36198273), the authors should assess the IFN signaling in inflamed tissue from the TBK1-deficient or kinase inactive backgrounds (e.g. *Ikke*^{-/-} *Tbk1*^{-/-} *Ripk1*^{wt/D138N}, *Ikke*^{-/-} *Tbk1*^{D135N/D135N} *Ripk1*^{wt/D138N}). This will provide some mechanistic understanding of how IKK ϵ modulates pro-inflammatory signaling in a TBK1-independent manner to drive spontaneous tissue inflammation. The authors should also use an unbiased approach (e.g. RNAseq or scRNAseq in the tissue) to further understand the IKK ϵ -dependency of cellular processes.
2. In most cases shown in the manuscript, RIPK1 kinase activity is critical for the development of disease in the *Ikke*^{-/-} *Tbk1*^{-/-} (or the kinase dead *Ikke*^{K38A/K38A} *Tbk1*^{D135N/D135N}) background. What is the status of RIPK1 phosphorylation (S166 and S321) in the tissue of inflamed vs uninflamed genotypes? And are there differences in phosphorylation and the ability to undergo cell death between different cell types, with one particular cell type having higher RIPK1 phosphorylation to drive cell death and

inflammation? The in vivo results seem to suggest that combined loss of TBK1 and IKKε in epithelial and myeloid cells (highly competent for pyroptosis), but not keratinocytes or liver cells (relatively resistant to pyroptosis), leads to RIPK1-dependent inflammation and cell death. Checking for RIPK1 phosphorylation will further define the RIPK1-dependent or -independent phenotypes.

3. What is the status of NF-κB (pIKBa) in TBK1-deficient myeloid vs non-myeloid cells? Does loss of IKKε further enhance NF-κB signaling in TBK1-deficient cells equally in both myeloid and non-myeloid cells?

4. The phenotypic rescue of *IkkeK38A/K38A Tbk1fl/D135N Cx3cr1-Crewt/tg* with *Il1.18.33r/-* is an interesting finding. What is the relationship between RIPK1 kinase activity and the production of these cytokines? Since *Il1.18.33r/-* can rescue both RIPK1-dependent phenotypes such as splenomegaly, granulocytosis, and monocytosis (Fig. 7), but also the RIPK1-independent T cell activation (Fig. 1G and Fig. 7C), it is important to assess the RIPK1 status in *IkkeK38A/K38A Tbk1fl/D135N Cx3cr1-Crewt/tg x Il1.18.33r/-* and cytokine levels in *IkkeK38A/K38A Tbk1D135N/D135N Ripk1D138N/D138N*, with their respective controls.

Minor comments

1. Quantification would help strengthen the manuscript in Fig. 3, Fig. 4, Fig. 5, Fig.7 and the corresponding supplemental figures.

Reviewer #3 (Remarks to the Author):

This study presented clear data on the consequence of simultaneous inhibition of TBK1 and IKKε kinase activities on embryonic viability and the inflammatory state of adult mice. Phenotypic characterization of the skin, liver and intestine that were conducted in many independent lines of mice provide compelling evidence for RIPK1 kinase-dependent and independent mechanisms of systemic inflammation upon inhibition of TBK1 and IKKε kinase activity. The paper could, however, be strengthened further by addressing the following points:

Major concerns:

1. The authors state that their results revealed 'previously unappreciated function of IKKε in compensating for the loss of TBK1 kinase activity' (line 81). While it is true that they have indeed shown functional redundancies between IKKε and TBK1 in maintaining homeostasis across multiple mouse tissues, work from others (Taft et al., 2021, ref #27) had already demonstrated a compensatory role for IKKε in the context of anti-viral type I interferon response in humans with TBK1 deficiency. This should be acknowledged in the description of the results.

2. Splenomegaly is reported in mice that are double knockouts or kinase dead (KD) for TBK1 or IKKε with heterozygous expression of RIPK1-D138N (Fig 1). The increased percentage of neutrophils found in these mice does not translate to neutrophil numbers (Fig 1F) and no considerable differences in lymphocytes are reported (Ex Data Fig 1). So what accounts for the enlarged size of their spleens— edema, stromal expansion?

3. The authors show clear features of heightened inflammation, such as splenomegaly, immune infiltration of the liver and ileum and hypotrophy of goblet cells, in mice with inhibition of TBK1 and IKKe expression or kinase activity, which are not or are partially rescued by homozygous expression of KD variant of RIPK1. Which RIPK1 kinase-independent function of TBK1 and IKKe is contributing to this phenotype?
4. Following treatment of WT BMDMs with MRT67307, the amount of IL-1b that is reported to be induced in Fig 6B and 6C have 1000x fold difference (ex- after WT cells are incubated with 10uM of the inhibitor for 20hrs in both experiments, ~0.25pg/ml of IL-1b is detected in Fig 6B whereas ~0.2ng/ml of IL-1b is shown in Fig 6C). Why the discrepancy?
5. Are elevated levels of IL-1b, IL18 or IL-33 detected in the serum of mice with combined deficiency or kinase activity inhibition of TBK1 and IKKe (either systemically or in the myeloid compartment only)? If so, is it dependent on RIPK1 kinase activity? Is there evidence for pyroptotic cell death, which is linked with the release of the IL-1 family of cytokines?
6. Would homozygous inhibition of RIPK1 kinase activity rescue IL-1b phenotype if BMDMs were to be derived from TBK1f/KD IKKeKD/KD Cx3cr1-Cre+/- mice? As pointed out by the authors, the inhibitors used in Fig 6 are efficient at targeting TBK1 activity but less potent towards IKKe kinase activity. Thus, the IL-1b response detected in WT BMDMs treated with the inhibitors could be milder than the one in BMDMs derived from myeloid-specific kinase dead mice. As such, tests to see if homozygous inhibition of RIPK1 kinase activity could reverse IL-1b or cell death could be misleading.
7. Missing validation showing absence of expression of the IL-1-, IL-18- and IL-33 receptor in Il1r-/-Il18r-/-IL33r-/- mice.
8. Missing indications of statistical significance in some quantifications throughout manuscript. For instance, Fig 7B, C.
9. Model systems are absolutely paramount in science. They naturally have limitations. Mice null for TBK1 die embryonically. Humans null for TBK1 are in their 30s now, some are still children. Thus, it is important to discuss these differences in detail, and title the paper such that it is obvious all was done in mice. Same needs to be followed in the abstract, and in each section of the text (this is paramount as reader needs to be reminded and frankly increasingly AI will be reading these texts, and we need to be precise, so that the body of discovery can be appropriately categorized, according to species specificities, which there are plenty of in the case of TBK1/IKKe biology).

Minor comments:

1. To strictly evaluate the impact of having a catalytically inactive TBK1 or IKKe proteins, it is important to confirm that the overall expression of the proteins isn't significantly affected. Although the authors conclude this is the case (line 110), looking at Ex Data Fig 1C lane #11, both the TBK1 and IKKe levels appear to be expressed much less than the control group.
2. p-IRF3 levels post LPS stimulation are quite low (Ex Fig 1C, lane 1-3) making it difficult to evaluate the effect of deleting or expressing catalytically inactive forms of TBK1 and IKKe

3. Extended Data Fig 1F- legend for panel 2 requires correction. It doesn't match what is shown.

We thank the reviewers for their thorough assessment of our work and their insightful comments, which have helped us further improve our manuscript. In our point-by-point response to the reviewers' comments below, we explain in detail how each point was addressed in the revised manuscript. In addition, we have re-organized the figures of the manuscript to make it easier for the readers to follow the flow of the paper. We hope that with the additional data and the new structure of the figures, the reviewers and the editors will now find our paper suitable for publication.

REVIEWER COMMENTS

Reviewer #1 (Remarks to the Author):

This is an elegant study that builds on previous observations from humans and mice that demonstrate that TBK1 has a critical role in preventing TNFR1-mediated and RIPK1-dependent cell death. The authors generated mice with the combined loss of IKK ϵ and TBK1 which causes embryonic lethality and evaluate the role of RIPK1 dependent and independent mechanisms in regulating embryonic development and tissue homeostasis.

RIPK1 kinase activity inhibition strongly suppressed, but could not fully prevent, the pathologies induced by combined inhibition of TBK1 and IKK ϵ , showing that TBK1/IKK ϵ also inhibit cell death and inflammation driven by RIPK1-independent mechanisms. In addition, the authors also reveal unique cell-type specific function of these kinases. TBK1 and IKK ϵ have an important role in myeloid cells and intestinal epithelial cells, where they limit RIPK1-mediated cell death and inflammation, but are dispensable in liver parenchymal cells and keratinocytes. Overall this is an extensive and comprehensive rigorous evaluation of combined loss of these two IKK related kinases and the studies are very clear and compelling.

We greatly appreciate the reviewer's comments on the elegant design and the rigorous and compelling nature of our studies and we thank them for their positive assessment of our work.

Reviewer #2 (Remarks to the Author):

In the article by Eren et al., the authors use several genetic crosses to examine the functional redundancies and relative roles of the related kinases TBK1 and IKK ϵ in cellular homeostasis. As the authors note, the cross-regulation between TBK1 and RIPK1 has been extensively studied to characterize the role of TBK1 in TNFR and IFN signaling, and the authors instead focus on characterizing TBK1's redundancies with the poorly studied IKK ϵ kinase in regulating the homeostasis of RIPK1 signaling in development, cell death, and inflammation. The authors also show a key role for the IL-1 family of cytokines in driving the RIPK1-mediated pathology, suggesting potential therapeutic targets for RIPK1-dependent pathologies. While the manuscript is primarily observational, the phenotypes are of interest, and the findings could be improved by additional mechanistic characterization.

Major comments

1. Given the critical role for TBK1 in IFN signaling to drive auto-inflammatory diseases and the spontaneous development of RIPK1-dependent disease in the context of caspase-8 deficiency (PMID: 29666472, PMID: 31827281, PMID: 31511692, PMID: 36198273), the authors should assess the IFN signaling in inflamed tissue from the TBK1-deficient or kinase

inactive backgrounds (e.g. *Ikke*^{-/-} *Tbk1*^{-/-} *Ripk1*^{wt/D138N}, *Ikke*^{-/-} *Tbk1*^{D135N/D135N} *Ripk1*^{wt/D138N}). This will provide some mechanistic understanding of how IKKε modulates pro-inflammatory signaling in a TBK1-independent manner to drive spontaneous tissue inflammation. The authors should also use an unbiased approach (e.g. RNAseq or scRNAseq in the tissue) to further understand the IKKε-dependency of cellular processes.

We thank the reviewer for their suggestion to assess whether IFN responses could be upregulated in the inflamed tissues of *Ikke*^{-/-} *Tbk1*^{-/-} *Ripk1*^{wt/D138N} and *Ikke*^{-/-} *Tbk1*^{-/-} *Ripk1*^{D138N/D138N} mice. We measured the expression of *Ifnb* and of three classical IFN stimulated genes (ISGs), namely *Oasl1*, *Ifit1* and *Ifi44* in the liver and small intestine of mice lacking TBK1 and IKKε expression or kinase activity in a homozygous or heterozygous RIPK1 kinase deficient background. As shown in Figure 1 for reviewers below, we did not detect considerably increased expression of *Ifnb* or of ISGs in RNA isolated from liver or small intestine of mice lacking TBK1 and IKKε expression or kinase activity compared to control animals. These results suggest that type I IFN responses are unlikely to play a role in the inflammatory pathology of these mice. This is not surprising, as we would not expect to find elevated type I IFN signaling in the absence of TBK1 and IKKε, which are required for type I IFN signaling activation downstream of multiple receptors.

Figure 1 for reviewers. Analysis of the expression of IFNβ and selected ISGs in mice lacking TBK1 and IKKε or their catalytic activity.

2. In most cases shown in the manuscript, RIPK1 kinase activity is critical for the development of disease in the *Ikke*^{-/-} *Tbk1*^{-/-} (or the kinase dead *Ikke*^{K38A/K38A} *Tbk1*^{D135N/D135N}) background. What is the status of RIPK1 phosphorylation (S166 and S321) in the tissue of inflamed vs uninfamed genotypes? And are there differences in phosphorylation and the ability to undergo cell death between different cell types, with one particular cell type having higher RIPK1 phosphorylation to drive cell death and inflammation? The in vivo results seem to suggest that combined loss of TBK1 and IKK ϵ in epithelial and myeloid cells (highly competent for pyroptosis), but not keratinocytes or liver cells (relatively resistant to pyroptosis), leads to RIPK1-dependent inflammation and cell death. Checking for RIPK1 phosphorylation will further define the RIPK1-dependent or -independent phenotypes.

Our conclusion that combined knockout or kinase deficiency of IKK ϵ and TBK1 causes cell death and inflammation mediated primarily by RIPK1 kinase activity is supported by our genetic experiments as well as by our in vitro studies using kinase inhibitors. The reviewer correctly points out that our genetic studies showed that combined inhibition of IKK ϵ and TBK1 exerted cell specific effects: in myeloid cells and intestinal epithelial cells it induced RIPK1-mediated cell death and inflammation, whereas in keratinocytes and liver parenchymal cells it did not cause cell death or inflammation. Importantly, inhibition of RIPK1 kinase activity could fully prevent cell death and inflammation in the intestinal epithelial-specific (*Ikke*^{K38A/K38A} *Tbk1*^{FL/D135N} *Vil1*-Cre) and the myeloid cell-specific (*Ikke*^{K38A/K38A} *Tbk1*^{FL/D135N} *Cx3cr1*-Cre) mice, demonstrating that RIPK1 catalytic activity drives cell death and inflammation in these models. Based on these clear genetic results, we would expect that RIPK1 should be activated and autophosphorylated on S166 in intestinal epithelial cells and myeloid cells but not in liver parenchymal cells and keratinocytes. We have made a considerable effort to establish immunohistochemical staining for phosphorylated RIPK1 S166, however, unfortunately, we failed to establish a protocol that works and gives reliable results. For this reason, we are not able to provide data assessing RIPK1 phosphorylation on S166 in the tissues of these mice. However, although we agree with the reviewer that it would be nice to show such stainings to complement our genetic results, we believe that our current dataset based on clear genetic findings fully supports our conclusions that RIPK1 kinase activity drives cell death and inflammation in response to IKK ϵ +TBK1 inhibition in intestinal epithelial and myeloid cells. Regarding RIPK1 S321, we would like to point out that phosphorylation on this site was shown to be mediated by MK2 and have a relatively mild inhibitory effect on RIPK1 activation. We are not sure what is the rationale of the reviewer about why S321 phosphorylation might be important in our model. Nevertheless, we have also attempted to assess S321 phosphorylation in the tissues of the mice but unfortunately also these immunostainings did not work.

As a last resort to assess phosphorylation on S166, we used protein lysates from livers of *Ikke*^{K38A/K38A} *Tbk1*^{D135N/D135N} *Ripk1*^{wt/D138N} and *Ikke*^{K38A/K38A} *Tbk1*^{D135N/D135N} *Ripk1*^{D138N/D138N} mice, from which we had available frozen liver tissue. Ripk1-kinase activity can induce caspase-8-dependent apoptosis (Polykratis et al., 2014). Our immunohistochemical analysis of liver tissue sections of *Ikke*^{K38A/K38A} *Tbk1*^{D135N/D135N} *Ripk1*^{wt/D138N} and *Ikke*^{K38A/K38A} *Tbk1*^{D135N/D135N} *Ripk1*^{D138N/D138N} mice revealed increased numbers of cell stained for cleaved caspase 8. Importantly, the liver sections of *Ikke*^{K38A/K38A} *Tbk1*^{D135N/D135N} *Ripk1*^{D138N/D138N} mice displayed less cleaved caspase 8 staining in comparison to the liver sections of *Ikke*^{K38A/K38A} *Tbk1*^{D135N/D135N} *Ripk1*^{wt/D138N} mice. These data indicated that RIPK1 kinase activity promotes caspase-8 activation in the liver of *Ikke*^{K38A/K38A} *Tbk1*^{D135N/D135N} *Ripk1*^{wt/D138N} mice. Therefore, we examined RIPK1 phosphorylation on S166 in liver lysates of *Ripk1*^{D138N/D138N}, *Ikke*^{K38A/K38A}

Tbk1^{D135N/D135N} *Ripk1*^{wt/D138N} and *Ikke*^{K38A/K38A} *Tbk1*^{D135N/D135N} *Ripk1*^{D138N/D138N} mice. As a positive control for these immunoblots we used lysates from lung fibroblasts from wild type mice stimulated with TNF and the SMAC mimetic compound birinapant. As shown in Figure 2 for reviewers below, we did not detect RIPK1 phosphorylation at S166 in the liver lysates from *Ikke*^{K38A/K38A} *Tbk1*^{D135N/D135N} *Ripk1*^{wt/D138N} mice. However, it is difficult to draw strong conclusions from this results, as it could be that the levels of RIPK1 phosphorylation in the heterozygous RIPK1 kinase dead genetic background are below the detection limit of this assay. For this reason, we did not include these results in the manuscript. Collectively, while we regret that we were not able to obtain data on RIPK1 phosphorylation in the tissues of our mice, we hope the reviewer will agree with us that our genetic studies clearly demonstrate the important role of RIPK1 kinase activity in driving cell death and inflammation in response to combined loss of inhibition of IKK ϵ and TBK1 in mice.

Figure 2 for reviewers. Immunoblot analysis of pS166 RIPK1 phosphorylation in liver lysates from *Ikke*^{K38A/K38A} *Tbk1*^{D135N/D135N} *Ripk1*^{wt/D138N} mice. Liver lysates from 8-13 week-old mice with the indicated genotypes with the indicated antibodies against indicated proteins. Cell lysates from WT lung fibroblasts treated with TNF(20ng/ml) and Birinapant, a SMAC mimetic, (1 μ M) for 3 hours were loaded as a control.

3. What is the status of NF- κ B (pIKBa) in TBK1-deficient myeloid vs non-myeloid cells? Does loss of IKK ϵ further enhance NF- κ B signaling in TBK1-deficient cells equally in both myeloid and non-myeloid cells?

We thank the reviewer for raising this question. We have indeed looked into the possible role of TBK1 and IKK ϵ in regulating TNF-induced NF- κ B signaling. Specifically, we stimulated lung fibroblasts and BMDMs from mice expressing kinase inactive TBK1, IKK ϵ and RIPK1 and compared their NF- κ B activation to wild type cells. We and others previously showed that inhibition of RIPK1 kinase activity in cells from *Ripk1*^{D138N/D138N} mice did not affect activation of NF- κ B in response to TNF stimulation (Newton et al., 2014; Polykratis et al., 2014), therefore

any difference in NF- κ B activation in cells expressing kinase inactive TBK1, IKK ϵ and RIPK1 compared to wild type cells could be attributed to TBK1 and IKK ϵ inhibition. To assess the activation of NF- κ B we used immunoblot against total I κ B α , which is rapidly degraded after stimulation and reappears within an hour due to the NF- κ B-dependent transcription of the I κ B α mRNA. We find that this is a robust assay for measuring NF- κ B activation, whereas the detection of phosphorylated I κ B α is often unreliable in our hands. We found that both lung fibroblasts and BMDMs from *Ikke*^{K38A/K38A} *Tbk1*^{D135N/D135N} *Ripk1*^{D138N/D138N} mice displayed similar I κ B α degradation pattern compared to wild type cells in response to TNF stimulation (Rev. Figure 3). Therefore, our experiments showed that the lack of TBK and IKK ϵ kinase activity did not considerably affect TNF-induced activation of NF- κ B.

Figure 3 for reviewers. The combined lack of TBK1 and IKK ϵ kinase activities did not affect NF- κ B signaling downstream of TNF in lung fibroblasts (A) and BMDMs (B). Lung fibroblasts and BMDMs from WT and *Ikke*^{K38A/K38A} *Tbk1*^{D135N/D135N} *Ripk1*^{D138N/D138N} mice were treated with TNF (20ng/ml) for the indicated time (minutes). Cell lysates were blotted with antibodies against the indicated proteins.

4. The phenotypic rescue of *Ikke*^{K38A/K38A} *Tbk1*^{fl/D135N} *Cx3cr1*-*Cre*^{wt}/*tg* with *Il1.18.33r*^{-/-} is an interesting finding. What is the relationship between RIPK1 kinase activity and the production of these cytokines? Since *Il1.18.33r*^{-/-} can rescue both RIPK1-dependent phenotypes such as splenomegaly, granulocytosis, and monocytosis (Fig. 7), but also the RIPK1-independent T cell activation (Fig. 1G and Fig. 7C), it is important to assess the RIPK1 status in *Ikke*^{K38A/K38A} *Tbk1*^{fl/D135N} *Cx3cr1*-*Cre*^{wt}/*tg* x *Il1.18.33r*^{-/-} and cytokine levels in *Ikke*^{K38A/K38A} *Tbk1*^{D135N/D135N} *Ripk1*^{D138N/D138N}, with their respective controls.

Here we would like to clarify that the rescue achieved by *Il1.18.33r*^{-/-} is very similar to that offered by inhibition of RIPK1 kinase activity. The reviewer mentions that T cell activation is independent of RIPK1 but depends on IL-1R family signaling, however, this is a misunderstanding caused by comparing data from the *Ikke*^{K38A/K38A} *Tbk1*^{D135N/D135N} *Ripk1*^{D138N/D138N} mice that express kinase inactive IKK ϵ , TBK1 and RIPK1 in all cells (shown in Figure 1), with data from with *Ikke*^{K38A/K38A} *Tbk1*^{fl/D135N} *Cx3cr1*^{Cre/wt} mice where IKK ϵ activity is missing systemically but TBK1 kinase activity is inhibited specifically in myeloid cells (shown in Figure 7). The data in Figure 7 need to be compared with the data in Figure 2, as in these figures we examined the myeloid cell-specific function of TBK1. Specifically, we assessed the

role of the IL-1R family in the phenotype caused by myeloid cell specific inhibition of TBK1 kinase activity together with systemic inhibition of IKK ϵ (*Ikke*^{K38A/K38A} *Tbk1*^{fl/D135N} *Cx3cr1*^{Cre/wt} \times *Il1.18.33r*^{-/-}) and found it strongly suppressed splenomegaly, granulocytosis and T cell activation (Figure 7). Inhibition of RIPK1 kinase activity rescued the phenotype in the same model (*Ikke*^{K38A/K38A} *Tbk1*^{fl/D135N} *Cx3cr1*^{Cre/wt} \times *Ripk1*^{D138N/D138N}), including splenomegaly, granulocytosis and T cell activation (Figure 2). Therefore, there is no discrepancy between the results obtained by inhibition of RIPK1 kinase activity or by knocking out IL1R, IL18R and IL33R, as in both cases the phenotype is rescued.

The reviewer also wonders what is the relationship between RIPK1 kinase activity and the production of IL-1 family cytokines. Our genetic data together with our in vitro mechanistic studies presented in Figure 6 provided experimental data supporting that RIPK1 drives the death of myeloid cells when IKK ϵ and TBK1 are inhibited, which triggers the production of IL-1 family cytokines including IL-1 β . As discussed above in response to the second point raised by the reviewer, assessing the phosphorylation of RIPK1 in tissue sections has been highly problematic in our hands. We have not been able to detect phosphorylation of RIPK1 in tissue sections in these mice and we also failed to detect RIPK1 phosphorylation in western blots of protein extracts from liver tissues. We should stress here that we can readily detect RIPK1 phosphorylation in cell cultures stimulated by appropriate stimuli, which we are using as our positive control in our experiments that failed to detect RIPK1 phosphorylation in tissue extracts. The most likely explanation for this is that the amount of cells with activated RIPK1 in vivo in our chronic models is low resulting in phospho-RIPK1 signal below the detection threshold, taking also into account that blots from whole tissues are not as clean as blots from cultured cell extracts. We are of course aware of publications by other groups presenting such stainings from tissues, however, although we have been using the same antibodies and protocols we have not been able to obtain reliable results. Nevertheless, our genetic data are consistent with an important role of RIPK1 kinase activity and fully support our conclusions.

Minor comments

1. Quantification would help strengthen the manuscript in Fig. 3, Fig. 4, Fig. 5, Fig.7 and the corresponding supplemental figures.

We appreciate the reviewer's suggestion that quantification could strengthen the immunostaining results. Following the advice of the reviewer, we have quantified the presence of cells with cleaved caspase-3 staining in intestinal tissues from the different mouse lines and include the results in the relevant figures of the revised manuscript (Fig.5c and Extended Data Fig.3c,d). We agree that these results provide additional support to the presented representative images. Regarding the other immunostainings, we believe that quantification will not add important information. Specifically, the CD45 immunostainings in intestinal sections reflect the infiltration of immune cells, which is already incorporated in the histological scoring of these sections that is presented in the respective figures. In the liver, the outcome of the CC8 immunostaining is binary, namely, the livers of the affected mice show CC8+ cells while the rescued and control mice do not contain any CC8+ cells. In this case, quantification will not provide additional information on the phenotype. Similarly, CD45 immunostainings in liver tissue also revealed a binary outcome, namely the livers of the affected mice have strong infiltration of CD45+ cells in the periportal areas, which are completely lacking in control and rescued mice that show only the presence of scattered single cells in the liver parenchyma,

presumably mainly Kupffer cells. In these cases, we believe that showing representative images is sufficient to support the conclusions. In order to show that the images we present in the figures of the manuscript are representative, we have prepared a collage of immunohistochemical stainings from three different mice from each genotype, which is presented in Figure 4 for reviewers below and is also included in the source data file accompanying the manuscript. This compilation visually captures the variations in staining patterns and supports the observations we are discussing in the manuscript.

CD45 staining collage of liver sections

Cleaved caspase 8 staining collage of liver sections

CD45 staining collage of small intestine sections

CD45 staining collage of colon sections

CD45 staining collage of intestine sections

Figure 4 for reviewers. Collage of representative pictures from tissue sections immunostained with the indicated antibodies. Tissues were collected from 8-13 weeks old mice with indicated genotypes. Each picture shows a representative image from an individual mouse with the specific genotype.

Reviewer #3 (Remarks to the Author):

This study presented clear data on the consequence of simultaneous inhibition of TBK1 and IKKe kinase activities on embryonic viability and the inflammatory state of adult mice. Phenotypic characterization of the skin, liver and intestine that were conducted in many independent lines of mice provide compelling evidence for RIPK1 kinase-dependent and independent mechanisms of systemic inflammation upon inhibition of TBK1 and IKKe kinase activity. The paper could, however, be strengthened further by addressing the following points:

Major concerns:

1. The authors state that their results revealed ‘previously unappreciated function of IKKe in compensating for the loss of TBK1 kinase activity’ (line 81). While it is true that they have indeed shown functional redundancies between IKKe and TBK1 in maintaining homeostasis across multiple mouse tissues, work from others (Taft et al., 2021, ref #27) had already demonstrated a compensatory role for IKKe in the context of anti-viral type I interferon

response in humans with TBK1 deficiency. This should be acknowledged in the description of the results.

We thank the reviewer for bringing this point to our attention. The statement mentioned by the reviewer meant to specifically refer to the role of IKK ϵ in compensating for the loss of TBK1 kinase activity to prevent RIPK1-dependent cell death and inflammation. While we believe that when seen in context this statement gave the intended meaning, we agree with the reviewer that, when taken out of context, this statement was ambiguous and could be misinterpreted to imply that it includes also the compensatory role of IKK ϵ in antiviral type I IFN responses. We have now re-phrased this statement to write: *“Our results revealed a previously unappreciated function of IKK ϵ in compensating for the loss of TBK1 kinase activity to prevent RIPK1-dependent cell death and inflammation, which is important for the maintenance of tissue homeostasis.”* We also agree that previous studies showed that IKK ϵ compensates for the loss of TBK1 in type I IFN signaling and acknowledged this function in the introduction of our manuscript stating that: *“IKK ϵ is homologous to TBK1 and was also shown to contribute to IFN production, with the two kinases exhibiting functional redundancy in anti-viral responses (Matsui et al., 2006; Taft et al., 2021)”*.

2. Splenomegaly is reported in mice that are double knockouts or kinase dead (KD) for TBK1 or IKK ϵ with heterozygous expression of RIPK1-D138N (Fig 1). The increased percentage of neutrophils found in these mice does not translate to neutrophil numbers (Fig 1F) and no considerable differences in lymphocytes are reported (Ex Data Fig 1). So what accounts for the enlarged size of their spleens— edema, stromal expansion?

It is indeed the case that the spleen size results are more clear than the neutrophil counts. However, most *Ikke*^{K38A/K38A} *Tbk1*^{D135N/D135N} *Ripk1*^{D138N/WT} mice have increased numbers of neutrophils compared to *Ikke*^{K38A/K38A} *Tbk1*^{D135N/D135N} *Ripk1*^{D138N/D138N} mice, although the variation in absolute numbers is relatively high. We have revised the graph in Figure 1F by dividing the x-axis into two segments to make it easier for the reader to appreciate the differences. We cannot exclude that additional factors may also contribute to the more clear differences in spleen weight /body weight ratio compared to absolute neutrophil numbers.

3. The authors show clear features of heightened inflammation, such as splenomegaly, immune infiltration of the liver and ileum and hypotrophy of goblet cells, in mice with inhibition of TBK1 and IKK ϵ expression or kinase activity, which are not or are partially rescued by homozygous expression of KD variant of RIPK1. Which RIPK1 kinase-independent function of TBK1 and IKK ϵ is contributing to this phenotype?

The reviewer raises an important question. Our studies showed that inhibition of RIPK1 kinase activity could rescue most, but not all aspects of the complex inflammatory pathology developing in mice with combined systemic deficiency of IKK ϵ and TBK1, demonstrating that the non-canonical IKKs also display RIPK1-independent functions driving inflammation in vivo. This RIPK1-independent pathology could be related to the multiple functions assigned to IKK ϵ and TBK1 in regulating different cellular processes. TBK1 and IKK ϵ are implicated in the regulation of multiple inflammatory signaling pathways (Heinz et al., 2020; Hemmi et al., 2004; Kagan et al., 2008; Seth et al., 2005; Yamamoto et al., 2003; Yoneyama et al., 2004; Zeng et al., 2010; Zhang et al., 2019; Zhao et al., 2019), metabolism (Zhao et al., 2018), mitochondrial

homeostasis (Heo et al., 2015; Lazarou et al., 2015; Richter et al., 2016) and autophagy (Matsumoto et al., 2015). While these functions of IKK ϵ and TBK1 could be involved in driving the RIPK1-independent pathology caused by combined inhibition of TBK1 and IKK ϵ , the specific underlying mechanisms remain elusive. Considering that answering this question will likely require extensive and lengthy studies and also taking into account the tremendous amount of work that we have already invested in our current study, we respectfully suggest that this goes beyond the scope of this manuscript and should be elucidated in future studies. Nevertheless, we have included a discussion on this topic in the discussion section of the manuscript (lines 420-428) to sensitize the reader to this aspect of our studies.

4. Following treatment of WT BMDMs with MRT67307, the amount of IL-1b that is reported to be induced in Fig 6B and 6C have 1000x fold difference (ex- after WT cells are incubated with 10uM of the inhibitor for 20hrs in both experiments, ~0.25pg/ml of IL-1b is detected in Fig 6B whereas ~0.2ng/ml of IL-1b is shown in Fig 6C). Why the discrepancy?

We thank the reviewer for spotting our mistake in the axis title. Indeed, there was a typo in **Figure 6B**, which should write ng/ml instead of pg/ml. We have now corrected this in the revised manuscript.

5. Are elevated levels of IL-1b, IL18 or IL-33 detected in the serum of mice with combined deficiency or kinase activity inhibition of TBK1 and IKK ϵ (either systemically or in the myeloid compartment only)? If so, is it dependent on RIPK1 kinase activity? Is there evidence for pyroptotic cell death, which is linked with the release of the IL-1 family of cytokines?

We have measured the serum levels of IL-1 β , IL-18 and IL-33 as well as IL-1 α , IP10 and IL-6 in the serum of mice lacking TBK1 and IKK ϵ kinase activity in myeloid cells (see **Figure 5 for reviewers** below). Although there was a trend towards increased serum levels of IL-1 β and IL-18 in *Ikke*^{K38A/K38A} *Tbk1*^{FL/D135N} *Cx3cr1-Cre*^{tg/wt} mice that was largely dependent on RIPK1 kinase activity, the large variability between individual mice makes it difficult to draw strong conclusions. Considering that the systemic inflammatory pathology is rather mild, it is perhaps not surprising that we could not find strong upregulation of cytokines in the circulation. We also do not have evidence for pyroptotic cell death being involved as we have not specifically assessed the role of pyroptosis mediators in our models. However, based on our cell culture experiment showing that inhibition of IKK ϵ and TBK1 in myeloid cells triggered RIPK1-mediated cell death resulting in IL-1 β production, we favor a model whereby RIPK1-mediated activation of cell death signaling (RIPK3-MLKL-dependent and caspase-8-mediated) in macrophages triggers the release of IL-1 cytokines. One possibility is that this might be driven by RIPK1-RIPK3-MLKL-dependent death of macrophages that results in indirect inflammasome activation and IL-1 cytokine release by inducing potassium efflux (Conos et al., 2017; Polykratis et al., 2019). Alternatively, caspase-8 activation could lead to RIPK3- and inflammasome-independent IL-1 β release, as shown previously (Bossaller et al., 2012). Although deciphering the detailed molecular mechanism by which IKK ϵ and TBK1 control IL-1 family cytokine release is an interesting question, considering the tremendous amount of work already included in the paper, we respectfully suggest that this is outside the scope of the current manuscript and should be the focus of future studies.

Figure 5 for reviewers: Cytokine levels in the serum of mice with myeloid-cell specific inhibition of IKK ϵ and TBK1. The indicated cytokines were measured in the serum of mice with the indicated genotypes using multiplex ELISA. Each dot represents an individual mouse.

6. Would homozygous inhibition of RIPK1 kinase activity rescue IL-1b phenotype if BMDMs were to be derived from TBK1f/KD IKK ϵ KD/KD Cx3cr1-Cre+/- mice? As pointed out by the authors, the inhibitors used in **Figure 6** are efficient at targeting TBK1 activity but less potent towards IKK ϵ kinase activity. Thus, the IL-1b response detected in WT BMDMs treated with the inhibitors could be milder than the one in BMDMs derived from myeloid-specific kinase dead mice. As such, tests to see if homozygous inhibition of RIPK1 kinase activity could reverse IL-1b or cell death could be misleading.

We have in fact tried the experiment suggested by the reviewer, namely to use BMDMs derived from *Ikke*^{K38A/K38A} *Tbk1*^{D138N/fl} *Cx3cr1-Cre* mice to assess the release of IL-1 β . Specifically, we isolated bone marrow cells from *Ikke*^{K38A/K38A} *Tbk1*^{FL/FL} *Cx3cr1-Cre*^{wt/wt}, *Ikke*^{K38A/K38A} *Tbk1*^{FL/D135N} *Cx3cr1-Cre*^{tg/wt}, and *Ikke*^{K38A/K38A} *Tbk1*^{FL/FL} *Cx3cr1-Cre*^{tg/wt} and differentiated them into macrophages in the presence of M-CSF. Subsequently, we treated these cells with different concentrations of LPS for 2 hours, prepared whole cell protein lysates and immunoblotted with antibodies against TBK1 to evaluate whether efficient TBK1 ablation was achieved in these cells, as well as for IKK ϵ as an internal control. As shown in **Figure 6 for reviewers** below, we found that the protein levels of TBK1 were only slightly decreased in *Ikke*^{K38A/K38A} *Tbk1*^{FL/FL} *Cx3cr1-Cre*^{wt/tg} compared to the cells lacking Cre expression (*Ikke*^{K38A/K38A} *Tbk1*^{FL/FL} *Cx3cr1-Cre*^{wt/wt}), revealing that the Cx3Cr1-Cre transgene did not achieve efficient deletion of TBK1 in BMDMs. This is not surprising, since it was previously reported that the Cx3cr1-Cre transgene does not efficiently delete loxP-flanked genes in BMDMs (Goldmann et al., 2013), although it is the most efficient and myeloid-cell specific Cre-expressing line for in vivo studies (Abram et al., 2014). Because of the inefficient Cre-mediated recombination, it was not possible to use these BMDMs to address how inhibition of TBK1 and IKK ϵ regulates IL-1 β release. For this reason, we decided to take an alternative approach, namely to use BMDMs from *Ikke*^{K38A/K38A} *Tbk1*^{D135N/D135N} *Ripk1*^{wt/D138N} mice, which lack IKK ϵ and TBK1 kinase activities and are heterozygous for the RIPK1 kinase inactive allele. To this end, we isolated bone marrow cells from *Ripk1*^{wt/wt}, *Ripk1*^{D138N/D138N}, *Tbk1*^{D135N/D135N} *Ripk1*^{wt/D138N}, *Tbk1*^{D135N/D135N} *Ripk1*^{D138N/D138N}, *Ikke*^{K38A/K38A} *Tbk1*^{D135N/D135N} *Ripk1*^{wt/D138N} and *Ikke*^{K38A/K38A} *Tbk1*^{D135N/D135N} *Ripk1*^{D138N/D138N} mice and differentiated them to macrophages in the presence of M-CSF. However, we observed that the macrophage yield from the bone marrow of *Ikke*^{K38A/K38A} *Tbk1*^{D135N/D135N} *Ripk1*^{wt/D138N} mice was strongly decreased compared to *Ikke*^{K38A/K38A} *Tbk1*^{D135N/D135N} *Ripk1*^{D138N/D138N} mice (**Figure 6 for reviewers**). In fact, we could not obtain enough BMDMs from *Ikke*^{K38A/K38A} *Tbk1*^{D135N/D135N} *Ripk1*^{wt/D138N} BM to perform functional studies. We believe that the reason behind the very poor yield of BMDMs from cells lacking IKK ϵ and TBK1 kinase activity is due to the fact that these cells undergo RIPK1 kinase-dependent cell death during differentiation. In summary, the poor deletion efficiency of Cx3cr1-Cre in BMDMs and the death of the bone marrow progenitors from *Ikke*^{K38A/K38A} *Tbk1*^{D135N/D135N}

Ripk1^{wt/D138N} mice made it impossible to assess IL-1 β release in cells with genetic inhibition of TBK1 combined with IKK ϵ .

However, we have used an alternative approach to address the issue pointed out by the reviewer, namely that BX795 and MRT67307 efficiently inhibit TBK1 but are less potent in inhibiting the kinase activity of IKK ϵ . To address this issue, we treated BMDMs from WT and *Ikke*^{K38A/K38A} mice with different concentrations of BX795 and MRT67307 to investigate whether genetic inhibition of IKK ϵ kinase activity will sensitize BMDMs to cell death and IL-1 β production at lower concentrations of BX795 and MRT67307 compared to wild type cells. The results from these experiments showed that *Ikke*^{K38A/K38A} BMDMs displayed increased cell death and produced higher amounts of IL-1 β compared to WT BMDMs when treated with low concentrations of BX795 and MRT67307 (**Figures 6E-F** of the revised manuscript). Importantly, inhibition of RIPK1 kinase activity suppressed cell death and prevented IL-1 β release in BMDMs from *Ikke*^{K38A/K38A} mice treated with different concentrations of BX795 and MRT67307 (**Figures 6E-F** of the revised manuscript). These results clearly showed that TBK1 and IKK ϵ act together to prevent RIPK1-kinase activity mediated cell death and the subsequent release of IL-1 β from macrophages. These findings are in line with our genetic data showing that combined deficiency of IL-1R, IL-18R, and IL-33R considerably suppressed the RIPK1-kinase activity driven systemic inflammatory phenotype of *Ikke*^{K38A/K38A} *Tbk1*^{FL/D135N} *Cx3cr1-Cre*^{wt/tg} mice (**Figure 7**). Taking all these in vitro and in vivo results together, we can conclude that the combined inhibition of IKK ϵ and TBK1 kinase activities causes RIPK1-kinase activity dependent IL-1 β release resulting in systemic inflammation.

Figure 6 for reviewers. (A) BMDMs from *Ikke*^{K38A/K38A} *Tbk1*^{FL/FL} *Cx3cr1-Cre*^{wt/wt}, *Ikke*^{K38A/K38A} *Tbk1*^{FL/D135N} *Cx3cr1-Cre*^{tg/wt}, and *Ikke*^{K38A/K38A} *Tbk1*^{FL/FL} *Cx3cr1-Cre*^{tg/wt} were treated with medium (M) or LPS (0.1ng/ml, 1ng/ml or 100ng/ml) for 2 hours. Whole cell protein lysates were blotted with the indicated antibodies. (B) Representative images of tissue culture dishes containing bone marrow cells from mice with the indicated genotypes that were differentiated in medium containing M-CSF for 6 days, and were subsequently fixed and stained with crystal violet.

7. Missing validation showing absence of expression of the IL-1-, IL-18- and IL-33 receptor in $Il1r^{-}/-Il18r^{-}/-Il33r^{-}/-$ mice.

In the previous version of our manuscript, to support that the $IL1.18.33R^{-}/-$ mice did not express IL-1R1, IL-18R and IL-33R we included functional data showing that bone marrow cells from these mice did not produce IL-6 in response to IL-1 β , IL-18 and IL-33 stimulation similarly to $MyD88^{-}/-$ cells (**Extended Data Fig 6B** of the originally submitted manuscript). However, we agree with the reviewer that assessment of the expression of the receptors would be critical to fully validate the mutant mice. We have now performed this experiment and provide new data demonstrating that BM cells from $IL1.18.33R^{-}/-$ mice lack expression of IL-1R1 and IL-33R (by FACS analysis) as well as IL-18R1 (by immunoblot analysis) (See **Figure 7 for reviewers below** and also new **Suppl. Fig. 4C-E**).

Figure 7 for reviewers. $IL1.18.33R^{-}/-$ mice lacks expression of IL-1R1, IL-18R1, and IL-33R. (A) Flow cytometry analysis of WT and $IL1.18.33R^{-}/-$ primary bone marrow cells for surface expression of IL-18R1 and IL-33R. (B) Western blot analysis of IL-1R1 in liver lysates from WT and $IL1.18.33R^{-}/-$ mice.

8. Missing indications of statistical significance in some quantifications throughout manuscript. For instance, Fig 7B, C.

We thank the reviewer for pointing out the missing statistical information in certain graphs. We have now added all missing statistics to the respective figures.

9. Model systems are absolutely paramount in science. They naturally have limitations. Mice null for TBK1 die embryonically. Humans null for TBK1 are in their 30s now, some are still children. Thus, it is important to discuss these differences in detail, and title the paper such that it is obvious all was done in mice. Same needs to be followed in the abstract, and in each section of the text (this is paramount as reader needs to be reminded and frankly increasingly AI will be reading these texts, and we need to be precise, so that the body of discovery can be appropriately categorized, according to species specificities, which there are plenty of in the case of TBK1/IKK ϵ biology).

We agree with the reviewer that species specificities need to be taken into account when studying biological processes and that precise description of the results is important to ensure

accurate reporting. In our study, it was clear that all experiments were performed in mice as this was already stated in the abstract and of course in every section in the results in the originally submitted manuscript, where we describe in detail the different mouse strains that were bred and analyzed. We also cited the study mentioned by the reviewer that describes the phenotype of TBK1-deficient humans in the introduction of our manuscript (lines 61 – 63, “*Importantly, mutations resulting in loss of TBK1 expression in humans caused systemic inflammatory pathologies that were effectively treated with anti-TNF antibodies (Taft et al., 2021)*”). We were therefore surprised to read the comment of the reviewer that our study might mislead the readers. In our opinion, our manuscript very clearly states that the work was performed in mice and we do not think it is necessary to repeatedly state in every sentence that the results were obtained in mice and mouse cells. Nevertheless, in order to further sensitize the readers to this topic, we have included the following text in the introduction of the revised manuscript (lines 324 – 329) “*Interestingly, a recent study reported that TBK1 deficiency did not cause embryonic lethality but resulted in TNF-mediated autoinflammatory pathology in human patients, suggesting that the role of IKK ϵ and TBK1 during embryonic development may be different in mice and humans, although these findings also need to take into account potential effects of genetic heterogeneity in the human population compared to inbred mouse strains*²⁷.”

We also feel the need to make a more general comment about this issue, which is indeed an important one. While there are several examples of gene knockouts that are embryonic lethal in mice but human patients lacking expression of these proteins are reported to be alive, this by no means provides proof that these proteins have different functions in mice and humans. These discrepancies can also arise from another reason, namely the effect of genetic background on the observed phenotypes. Studies in genetically modified mice are classically performed in inbred strains with well-defined genetic background, primarily in C57BL/6 mice. There are several examples in the literature that a gene knockout may cause embryonic lethality in a certain genetic background, but when the mice are backcrossed to another genetic background they are viable (e.g. (Doetschman, 1999; LeCouter et al., 1998; Sibilias and Wagner, 1995)). These studies suggest that modifier genes may profoundly influence the phenotype caused by a given gene deficiency or mutation resulting in a wide spectrum of phenotypes ranging from early embryonic lethality to a viable mouse sometimes reaching adulthood. Obviously, studies in human patients are restricted to the discovery and characterization of individuals that are born and display specific pathologies, and cannot assess the effect of a certain mutation in a well-controlled genetic study as in experimental models. Taking into account the tremendous genetic variation of the human population, it is therefore possible, perhaps even likely, that a certain gene deficiency that is reported not to cause embryonic lethality in certain individuals may be embryonic lethal in other individuals based on allelic variation of modifier genes. Specifically, in the case of TBK1, its role in regulating RIPK1-dependent cell death has been well-established in both mouse and human cells therefore this is conserved across the two species. Our study reveals a previously unappreciated contribution of IKK ϵ in cooperating with TBK1 to suppress RIPK1-dependent and -independent cell death and inflammation in mice. These findings in a model organism are relevant for better understanding the role of these kinases in humans and will be particularly valuable for the better design of studies using inhibitors of these kinases for the treatment of disease.

Minor comments:

1. To strictly evaluate the impact of having a catalytically inactive TBK1 or IKKε proteins, it is important to confirm that the overall expression of the proteins isn't significantly affected. Although the authors conclude this is the case (line 110), looking at Ex Data Fig 1C lane #11, both the TBK1 and IKKε levels appear to be expressed much less than the control group.

Indeed, the blots in lane 11 seem as if TBK1 and IKKε are expressed at lower levels, but this is not the case. As shown in the duplicate experiment included below in **Figure 8 for reviewers**, the protein expression levels of kinase inactive TBK1 and IKKε are similar to those of the wild type proteins. Moreover, our additional experiments assessing the activation of NF-κB in **Figure 2 for reviewers** above provide additional evidence that the kinase inactive proteins are expressed at similar levels as the wild type proteins. We have now replaced the blots in new **Fig. 1E** in the revised manuscript with these blots to avoid giving the false impression to the reader that kinase inactive IKKε and TBK1 may be expressed at lower levels.

Figure 8 for reviewers. The lack TBK1 and IKKε kinase activity does not affect the expression of TBK1 and IKKε, and blocks type I IFN production in response to LPS. (A) BMDMs were treated with LPS (100ng/ml) for 1 hour, and cell lysates were blotted for the indicated antibodies. (B) ELISA analysis of IFN-β protein level in LPS (100ng/ml) treated BMDMs from the indicated genotype of mice for 24 hours

2. p-IRF3 levels post LPS stimulation are quite low (Extended Data 1C, lane 1-3) making it difficult to evaluate the effect of deleting or expressing catalytically inactive forms of TBK1 and IKKε

Indeed, the p-IRF3 signal is low, however the blots are clean and clearly show that phosphorylation of IRF3 is impaired in the absence of TBK1 and IKKε activity. This can also be clearly seen in the duplicate experiment presented in **Figure 8 for reviewers** above.

3. Extended Data Fig 1F- legend for panel 2 requires correction. It doesn't match what is shown.

We thank the reviewer for spotting our mistake. We have corrected the legend in the revised manuscript.

References

- Abram, C.L., Roberge, G.L., Hu, Y., and Lowell, C.A. (2014). Comparative analysis of the efficiency and specificity of myeloid-Cre deleting strains using ROSA-EYFP reporter mice. *J Immunol Methods* *408*, 89-100.
- Bossaller, L., Chiang, P.I., Schmidt-Lauber, C., Ganesan, S., Kaiser, W.J., Rathinam, V.A., Mocarski, E.S., Subramanian, D., Green, D.R., Silverman, N., Fitzgerald, K.A., Marshak-Rothstein, A., and Latz, E. (2012). Cutting edge: FAS (CD95) mediates noncanonical IL-1beta and IL-18 maturation via caspase-8 in an RIP3-independent manner. *J Immunol* *189*, 5508-5512.
- Conos, S.A., Chen, K.W., De Nardo, D., Hara, H., Whitehead, L., Nunez, G., Masters, S.L., Murphy, J.M., Schroder, K., Vaux, D.L., Lawlor, K.E., Lindqvist, L.M., and Vince, J.E. (2017). Active MLKL triggers the NLRP3 inflammasome in a cell-intrinsic manner. *Proc Natl Acad Sci U S A* *114*, E961-E969.
- Doetschman, T. (1999). Interpretation of phenotype in genetically engineered mice. *Lab Anim Sci* *49*, 137-143.
- Goldmann, T., Wieghofer, P., Muller, P.F., Wolf, Y., Varol, D., Yona, S., Brendecke, S.M., Kierdorf, K., Staszewski, O., Datta, M., Luedde, T., Heikenwalder, M., Jung, S., and Prinz, M. (2013). A new type of microglia gene targeting shows TAK1 to be pivotal in CNS autoimmune inflammation. *Nat Neurosci* *16*, 1618-1626.
- Heinz, L.X., Lee, J., Kapoor, U., Kartnig, F., Sedlyarov, V., Papakostas, K., Cesar-Razquin, A., Essletzbichler, P., Goldmann, U., Stefanovic, A., Bigenzahn, J.W., Scorzoni, S., Pizzagalli, M.D., Bensimon, A., Muller, A.C., King, F.J., Li, J., Girardi, E., Mbow, M.L., Whitehurst, C.E., Rebsamen, M., and Superti-Furga, G. (2020). TASL is the SLC15A4-associated adaptor for IRF5 activation by TLR7-9. *Nature* *581*, 316-322.
- Hemmi, H., Takeuchi, O., Sato, S., Yamamoto, M., Kaisho, T., Sanjo, H., Kawai, T., Hoshino, K., Takeda, K., and Akira, S. (2004). The roles of two IkappaB kinase-related kinases in lipopolysaccharide and double stranded RNA signaling and viral infection. *J Exp Med* *199*, 1641-1650.
- Heo, J.M., Ordureau, A., Paulo, J.A., Rinehart, J., and Harper, J.W. (2015). The PINK1-PARKIN Mitochondrial Ubiquitylation Pathway Drives a Program of OPTN/NDP52 Recruitment and TBK1 Activation to Promote Mitophagy. *Mol Cell* *60*, 7-20.
- Kagan, J.C., Su, T., Horng, T., Chow, A., Akira, S., and Medzhitov, R. (2008). TRAM couples endocytosis of Toll-like receptor 4 to the induction of interferon-beta. *Nat Immunol* *9*, 361-368.
- Lazarou, M., Sliter, D.A., Kane, L.A., Sarraf, S.A., Wang, C., Burman, J.L., Sideris, D.P., Fogel, A.I., and Youle, R.J. (2015). The ubiquitin kinase PINK1 recruits autophagy receptors to induce mitophagy. *Nature* *524*, 309-314.
- LeCouter, J.E., Kablar, B., Whyte, P.F., Ying, C., and Rudnicki, M.A. (1998). Strain-dependent embryonic lethality in mice lacking the retinoblastoma-related p130 gene. *Development* *125*, 4669-4679.

- Matsui, K., Kumagai, Y., Kato, H., Sato, S., Kawagoe, T., Uematsu, S., Takeuchi, O., and Akira, S. (2006). Cutting edge: Role of TANK-binding kinase 1 and inducible I κ B kinase in IFN responses against viruses in innate immune cells. *J Immunol* 177, 5785-5789.
- Matsumoto, G., Shimogori, T., Hattori, N., and Nukina, N. (2015). TBK1 controls autophagosomal engulfment of polyubiquitinated mitochondria through p62/SQSTM1 phosphorylation. *Hum Mol Genet* 24, 4429-4442.
- Newton, K., Dugger, D.L., Wickliffe, K.E., Kapoor, N., de Almagro, M.C., Vucic, D., Komuves, L., Ferrando, R.E., French, D.M., Webster, J., Roose-Girma, M., Warming, S., and Dixit, V.M. (2014). Activity of protein kinase RIPK3 determines whether cells die by necroptosis or apoptosis. *Science* 343, 1357-1360.
- Polykratis, A., Hermance, N., Zelic, M., Roderick, J., Kim, C., Van, T.M., Lee, T.H., Chan, F.K.M., Pasparakis, M., and Kelliher, M.A. (2014). Cutting edge: RIPK1 Kinase inactive mice are viable and protected from TNF-induced necroptosis in vivo. *J Immunol* 193, 1539-1543.
- Polykratis, A., Martens, A., Eren, R.O., Shirasaki, Y., Yamagishi, M., Yamaguchi, Y., Uemura, S., Miura, M., Holzmann, B., Kollias, G., Armaka, M., van Loo, G., and Pasparakis, M. (2019). A20 prevents inflammasome-dependent arthritis by inhibiting macrophage necroptosis through its ZnF7 ubiquitin-binding domain. *Nat Cell Biol* 21, 731-742.
- Richter, B., Sliter, D.A., Herhaus, L., Stolz, A., Wang, C., Beli, P., Zaffagnini, G., Wild, P., Martens, S., Wagner, S.A., Youle, R.J., and Dikic, I. (2016). Phosphorylation of OPTN by TBK1 enhances its binding to Ub chains and promotes selective autophagy of damaged mitochondria. *Proc Natl Acad Sci U S A* 113, 4039-4044.
- Seth, R.B., Sun, L., Ea, C.K., and Chen, Z.J. (2005). Identification and characterization of MAVS, a mitochondrial antiviral signaling protein that activates NF- κ B and IRF 3. *Cell* 122, 669-682.
- Sibilia, M., and Wagner, E.F. (1995). Strain-dependent epithelial defects in mice lacking the EGF receptor. *Science* 269, 234-238.
- Taft, J., Markson, M., Legarda, D., Patel, R., Chan, M., Malle, L., Richardson, A., Gruber, C., Martin-Fernandez, M., Mancini, G.M.S., van Laar, J.A.M., van Pelt, P., Buta, S., Wokke, B.H.A., Sabli, I.K.D., Sancho-Shimizu, V., Chavan, P.P., Schnappauf, O., Khubchandani, R., Cuceoglu, M.K., Ozen, S., Kastner, D.L., Ting, A.T., Aksentijevich, I., Hollink, I., and Bogunovic, D. (2021). Human TBK1 deficiency leads to autoinflammation driven by TNF-induced cell death. *Cell* 184, 4447-4463 e4420.
- Yamamoto, M., Sato, S., Hemmi, H., Hoshino, K., Kaisho, T., Sanjo, H., Takeuchi, O., Sugiyama, M., Okabe, M., Takeda, K., and Akira, S. (2003). Role of adaptor TRIF in the MyD88-independent toll-like receptor signaling pathway. *Science* 301, 640-643.
- Yoneyama, M., Kikuchi, M., Natsukawa, T., Shinobu, N., Imaizumi, T., Miyagishi, M., Taira, K., Akira, S., and Fujita, T. (2004). The RNA helicase RIG-I has an essential function in double-stranded RNA-induced innate antiviral responses. *Nat Immunol* 5, 730-737.
- Zeng, W., Sun, L., Jiang, X., Chen, X., Hou, F., Adhikari, A., Xu, M., and Chen, Z.J. (2010). Reconstitution of the RIG-I pathway reveals a signaling role of unanchored polyubiquitin chains in innate immunity. *Cell* 141, 315-330.
- Zhang, C., Shang, G., Gui, X., Zhang, X., Bai, X.C., and Chen, Z.J. (2019). Structural basis of STING binding with and phosphorylation by TBK1. *Nature* 567, 394-398.
- Zhao, B., Du, F., Xu, P., Shu, C., Sankaran, B., Bell, S.L., Liu, M., Lei, Y., Gao, X., Fu, X., Zhu, F., Liu, Y., Laganowsky, A., Zheng, X., Ji, J.Y., West, A.P., Watson, R.O., and Li, P. (2019). A conserved PLPLRT/SD motif of STING mediates the recruitment and activation of TBK1. *Nature* 569, 718-722.

Zhao, P., Wong, K.I., Sun, X., Reilly, S.M., Uhm, M., Liao, Z., Skorobogatko, Y., and Saltiel, A.R. (2018). TBK1 at the Crossroads of Inflammation and Energy Homeostasis in Adipose Tissue. *Cell* 172, 731-743 e712.

REVIEWERS' COMMENTS

Reviewer #1 (Remarks to the Author):

The authors have provided a compelling series of additional data to support the overall model they propose. I have no further concerns and commend the authors on a thorough and interesting study.

Reviewer #2 (Remarks to the Author):

In their revision, the authors have adequately addressed my comments. I have no additional points.

Reviewer #3 (Remarks to the Author):

In the revised version of their manuscript, the authors have sufficiently addressed the concerns I had previously raised or explained why their existing systems do not reasonably allow for them to perform the suggested experiments.

I would recommend that the authors consider making the following amendments in the title and discussion of their findings:

1. Previous comments to clearly delineate the differences between human and mouse TBK1/IKKe biology, within the context of their findings, was not intended to diminish the implication of the current work, conducted in mouse models, on human biology. Rather, it was to appropriately categorize and accurately document settings in which the two systems have overlapping and distinct features. The comments provided to the authors neither stated that the manuscript is misleading nor was it requested that every sentence in the results section should repeatedly state that it was conducted in mice/mouse cells. The comment was meant to ensure that conclusions made, whenever possible, were contrasted with findings in humans to ensure contextualization of the results. In the revised manuscript, the authors have indeed made efforts to add sentences in the introduction and discussion that reflect some of these species-specific differences. I strongly suggest that authors similarly amend the title of the paper to indicate these findings were in mice so that is clear to the readers.

2. Mice with dual ablation of TBK1 and IKKe expression or kinase activity exhibit heightened inflammation, which is not or is partially rescued by homozygous expression of KD variant of RIPK1. In discussing the potential mechanisms underlying this pathology (line 420-428), the authors rightly bring up the likely involvement of RIPK1-independent functions mediated by TBK1 and IKKe. However, they overlook the possibility of RIPK1-mediated kinase-independent function (ex- in nucleating formation of various complexes) contributing to this phenotype. Inclusion of this possibility in the discussion would broaden possibilities for interpreting the results.

Minor comment:

- In Supp Fig 4C- please indicate frequency of IL-33R+ and IL-18R1+ cells.

REVIEWERS' COMMENTS

Reviewer #1 (Remarks to the Author):

The authors have provided a compelling series of additional data to support the overall model they propose. I have no further concerns and commend the authors on a thorough and interesting study.

We thank the reviewer for their feedback and comments that were valuable in improving the quality of our paper.

Reviewer #2 (Remarks to the Author):

In their revision, the authors have adequately addressed my comments. I have no additional points.

We thank the reviewer for their feedback and comments that were valuable in improving the quality of our paper.

Reviewer #3 (Remarks to the Author):

In the revised version of their manuscript, the authors have sufficiently addressed the concerns I had previously raised or explained why their existing systems do not reasonably allow for them to perform the suggested experiments.

I would recommend that the authors consider making the following amendments in the title and discussion of their findings:

1. Previous comments to clearly delineate the differences between human and mouse TBK1/IKKe biology, within the context of their findings, was not intended to diminish the implication of the current work, conducted in mouse models, on human biology. Rather, it was to appropriately categorize and accurately document settings in which the two systems have overlapping and distinct features. The comments provided to the authors neither stated that the manuscript is misleading nor was it requested that every sentence in the results section should repeatedly state that it was conducted in mice/mouse cells. The comment was meant to ensure that conclusions made, whenever possible, were contrasted with findings in humans to ensure contextualization of the results. In the revised manuscript, the authors have indeed made efforts to add sentences in the introduction and discussion that reflect some of these species-specific differences. I strongly suggest that authors similarly amend the title of the paper to indicate these findings were in mice so that is clear to the readers.

We thank the reviewer for their thorough assessment of our manuscript and their insightful comments. As we discussed in response to a similar comment of the reviewer in the last round of review, our manuscript clearly states that our experiments were performed in mice and mouse cells. This information is explicitly included in the abstract, the results section and in the discussion. With all respect to the reviewer's opinion, we do not consider it necessary to also state this in the title. This is also consistent with established practices in scientific literature. There are now hundreds of thousands of papers reporting results in mice, as GEMMs have become the main experimental system in biomedical research, however, in the vast majority of these papers the term 'mice' does not appear in the title. We therefore respectfully suggest that we do not have to state in the title of the paper that the work was performed in mice.

2. Mice with dual ablation of TBK1 and IKK ϵ expression or kinase activity exhibit heightened inflammation, which is not or is partially rescued by homozygous expression of KD variant of RIPK1. In discussing the potential mechanisms underlying this pathology (line 420-428), the authors rightly bring up the likely involvement of RIPK1-independent functions mediated by TBK1 and IKK ϵ . However, they overlook the possibility of RIPK1-mediated kinase-independent function (ex- in nucleating formation of various complexes) contributing to this phenotype. Inclusion of this possibility in the discussion would broaden possibilities for interpreting the results.

We thank the reviewer for this comment. To address this point, we have included in the discussion of the manuscript the following sentence: "*In addition to its kinase activity-dependent cell death-inducing functions, RIPK1 also acts as a scaffold to promote proinflammatory signalling, which could be implicated in driving inflammation in mice lacking TBK1 and IKK ϵ kinase activities.*"

Minor comment:

- In Supp Fig 4C- please indicate frequency of IL-33R+ and IL-18R1+ cells.

We thank the reviewer for pointing out this omission. This information is now included in the flow cytometry plots.